

# Burning conditions and transportation pathways determine biomass-burning aerosol properties in the Ascension Island marine boundary layer

Amie Dobracki[1], Ernie R. Lewis[1], Arthur J. Sedlacek III[1], Tyler Tatro[2], Maria A. Zawadowicz[1], Paquita Zuidema[2]

[1]Brookhaven National Laboratory, Upton, NY, USA
[2]Department of Atmospheric Sciences, Rosenstiel School, University of Miami, Miami, Florida, USA

*Correspondence to*: Amie Dobracki (adobracki@bnl.gov) and Paquita Zuidema (pzuidema@miami.edu)

**Abstract:**

African biomass-burning aerosol (BBA) in the southeast Atlantic Ocean (SEA) marine boundary layer (MBL) is an important contributor to Earth's radiation budget yet its representation remains poorly constrained in regional and global climate models. Data from the Layered Atlantic Smoke Interactions with Clouds (LASIC) field campaign on Ascension Island (-7.95˚N, -14.36˚E) detail how fire source regions (burning conditions and fuel type), transport pathways, and longer-term chemical processing affect the chemical, microphysical, and optical properties of the BBA in the remote MBL between June and September of 2017. Ten individual plume events characterize the seasonal evolution of BBA characteristics. Inefficient burning conditions, determined by the mass ratio of refractory black carbon to above-background carbon monoxide ($rBC:\Delta CO$), enhance organic- and sulfate-rich aerosol concentrations in June-July. In contrast, the heart of the burning season exhibited higher $rBC:\Delta CO$ values indicative of efficient burning conditions, correlating with more rBC-enriched BBA. Toward the end of the burning season, a mix of burning conditions results in increased variation of the BBA properties. The BBA transit to Ascension Island was predominantly through slow-moving pathways in the MBL and lower free troposphere (FT), facilitating prolonged chemical transformations through heterogeneous and aqueous phase processes. Heterogeneous oxidation can persist for up to 10 days, resulting in a considerable decrease in organic aerosol (OA) mass. OA to rBC mass ratios (OA:rBC) in the MBL between 2 and 5 contrast to higher values of 5 to 15 observed in the nearby FT. Conversely, early-season aqueous-phase processes primarily contributed to aerosol oxidation and some aerosol production, but not appreciable aerosol removal. These two chemical processes yield more light-absorbing BBA in the MBL than in the FT and explain the notably low scattering albedo at 530 nm ($SSA_{530}$) values (< 0.80) at Ascension Island. This study establishes a robust correlation between $SSA_{530}$ and OA:rBC across both MBL and FT, underscoring the dependency of optical properties on chemical composition. These findings highlight how the interplay between chemical composition and atmospheric processing can be improved in global and regional climate models. Questions remain on the mixing of aerosols with different pathway histories, and on what accounts for the doubling of the mass absorption coefficient in the boundary layer.



## 1. Introduction

Biomass burning injects 16-34 Tg of particulate matter into Earth's atmosphere each year, with Africa contributing over 50% of that amount (Akagi et al., 2011; Bond et al., 2013). BBA and the copious amounts of co-emitted gases such as carbon dioxide, carbon monoxide, and ozone, have major effects on air quality, climate change, human health, and Earth's geochemical cycles (Bowman et al., 2009, Bond et al., 2013). Biomass burning is a major global source of black carbon (BC), the second largest contributor to climate warming after carbon dioxide ($CO_2$). However, BC is a minor component of the total biomass burning aerosol (BBA), which also includes organic aerosol (OA), sulfate ($SO_4$), nitrate ($NO_3$), and ammonium ($NH_4$). The large impacts of BBA on direct aerosol radiative forcing and aerosol-cloud interactions, both of which contribute substantial uncertainty to Earth's radiative budget (IPCC, 2021), underlie the importance of detailed characterization of its chemical, microphysical, and optical properties.

A deck of persistent marine stratocumulus clouds offshore of southwestern Africa coexists with biomass burning in southern Africa from June to October of each year, creating a natural laboratory to investigate how long-range transported BBA interacts with low clouds far to the west of the continent (Zuidema et al., 2016; Adebiyi and Zuidema, 2018). Recent field campaigns have determined that the BBA is commonly transported across the southeast Atlantic Ocean (SEA) in both the free troposphere (FT) and the marine boundary layer (MBL) during these five months (Zuidema et al., 2018; Haywood et al., 2021; Redemann et al., 2021). The BBA in the FT is highly absorbing of shortwave radiation (Pistone et al., 2019; Denjean et al., 2020a; Taylor et al., 2020; Wu et al., 2020; Dobracki et al., 2023) with August-September mean single scatting albedo (the ratio of the aerosol scattering coefficient to the sum of the aerosol scattering and absorption coefficients) at wavelength 530 nm ($SSA_{530}$) values near 0.84 (Pistone et al., 2019; Wu et al., 2020, Dobracki et al., 2023). Measurements in the MBL at Ascension Island (-7.95˚N, -14.36˚E), a remote location in the tropical Atlantic, yield an even lower $SSA_{530}$, with June-August monthly-mean values near 0.80 (Zuidema et al., 2018; Che et al., 2022a). The lower $SSA_{530}$ at Ascension Island in the MBL compared to that in the FT above the island has not been previously explained (Barrett et al., 2022; Sedlacek et al., 2022).

Multiple field campaigns have collected spatially comprehensive data sets on aerosol properties across the SEA, yet substantial observation-model discrepancies for this region persist (Chylek et al., 2019; Hodzic et al., 2020; Mallet et al., 2020; Shinozuka et al., 2020; Brown et al., 2021; Doherty et al., 2022; Howes et al., 2023). Measured $SSA_{530}$ values in the FT and MBL across the SEA are much lower than those used in many global and regional climate models (Shinozuka et al., 2020; Mallet et al., 2020; Doherty et al., 2022). The aerosol over the SEA is highly oxidized (Hodzic et al., 2020; Dobracki et al., 2023), with measured molecular ratios of organic matter to organic carbon (OM:OC) near 2.25 in the FT in September, exceeding the value of 1.4 that is used in many climate models (Hodzic et al., 2020). Aerosol removal through chemical and photolytic processing is typically not represented in models (Hodzic et al., 2020), which contributes to the underestimation of OM:OC. Solar absorption in the FT over the SEA is also not well captured in climate models because of unrealistic aerosol chemical composition (Mallet et al., 2021), inaccurate representation of mixing states (Brown et al., 2021), and overestimation of the rate at which absorbing aerosol descends in the atmosphere, which also leads to underestimation in the aerosol loading



in the FT (Das et al., 2017). These studies demonstrate that more realistic representations of aerosol properties and processes in the SEA are required to yield accurate results.

Much of the research over the SEA has focused on BBA properties in the FT, with fewer studies characterizing and constraining BBA properties in the MBL. Here, data from the Layered Atlantic Smoke Interactions with Clouds (LASIC) field campaign on Ascension Island are used to examine how source regions and conditions (i.e., burning conditions and fuel type),

chemical processes, and transport pathways affect the chemical, microphysical, and optical properties of the BBA in the MBL between June and September of 2017. Results from LASIC are compared to those of Wu et al. (2020) and of Taylor et al. (2020), both of whom analyzed BBA properties during the Cloud-Aerosol Radiation Interaction and Forcing Year 2017 (CLARIFY) aircraft campaign in August-September 2017 (Haywood et al., 2021). The CLARIFY analysis, while based on only three weeks of data, covered times when BBA was present in only the FT, only the MBL, and both the FT and the MBL.

We also compare our results to those of Dobracki et al. (2023), who analyzed BBA properties in the FT near the African coast for September, 2016 during the ObseRvations of Aerosols above CLouds and their IntEractionS (ORACLES 2016-2018) campaign (Redemann et al., 2021). Moreover, we investigate whether the strong relationship between $SSA_{530}$ and the mass ratio of OA to BC (OA:BC) observed during ORACLES, documented in Dobracki et al. (2023), also applies to the BBA within the remote MBL.

We distinguish BBA characteristics that are primarily determined at the source, such as BC core diameter and above-background CO ($\Delta$CO) mixing ratio, from those that can change during long-range transport, such as chemical composition and size distribution, by examining back trajectories for selected time periods when BC mass concentrations in the MBL were sufficiently high. We then combine the back-trajectory estimates of BBA source locations with maps of fire density, surface relative humidity (RH), and land use to infer the burning conditions and fuel types at the fire sources. We address the following

questions:

1. *How do the chemical, microphysical, and optical properties of African BBA in the MBL at Ascension Island change between June and September?*
2. *How do transport pathways from Africa to Ascension Island impact the chemical, microphysical, and optical properties of African BBA in the MBL there?*
3. *How do differences in burning conditions and fuel type affect the chemical, microphysical, and optical properties of African BBA in the MBL at Ascension Island?*
4. *How do the chemical, microphysical, and optical properties of African BBA in the MBL at Ascension Island differ from those in the FT above?*
5. *Is there a clear relationship between $SSA_{530}$ and OA:BC for the MBL at Ascension Island similar to that reported by*
*Dobracki et al. (2023) for the FT closer to the coast?*

In Sect. 2, we introduce the LASIC field campaign, the sampling and instrumentation methods, the data products, and the data analysis techniques employed in this study. In Sect. 3.1, we identify 10 events with high BC mass concentrations, aggregated into three temporal regimes distinguished by burning conditions, fuel type, and the conserved tracers BC and $\Delta$CO.

In Sect. 3.2, we present the chemical, microphysical, and optical properties of the BBA in the Ascension Island MBL for the 10 events. Next, in Sect. 3.3, we discuss the aerosol transport pathways of the BBA to Ascension Island. In Sect. 4, we discuss



the evolution of BC properties (Sects. 4.1-4.2), oxidation processes, and transport pathways (Sects. 4.3-4.4 and the dependence of SSA$_{530}$ on BBA chemical properties (Sect. 4.5). Sect. 5 highlights a change in aerosol transport in early September (Sect. 5.1) and discusses remaining questions on enhanced aerosol absorption (Sect. 5.2). Lastly, in Sect. 6, we conclude with
a summary of our major findings.

## 2. Methods

### 2.1 Instrumentation and data products

The LASIC field campaign deployed the Department of Energy (DOE) Atmospheric Radiation Measurement (ARM) First Mobile Facility (AMF1) (Zuidema et al., 2018) on Ascension Island between June 2016 and October 2017 to measure
aerosol and cloud properties with the goal of constraining uncertainties in BBA aging processes and aerosol-cloud interactions. Here we focus on measurements collected from June to September of 2017, as aerosol composition was available only for those times. The utilized data products, quantities measured, and instrumentation suite are listed in Table 1.

Mass concentrations of OA, SO$_4$, NO$_3$, and NH$_4$ were measured with the quadrupole Aerosol Chemical Speciation Monitor (ACSM; Ng et al., 2011), which separates the ions of different elemental compositions at each mass-to-charge ratio
(m/z) to provide unit-mass resolution. The sampling efficiency of the LASIC ACSM was previously discussed in Barrett et al. (2022); however, a comparison (Appendix A) of the aerosol volume concentration calculated using the ACSM with that calculated from the size distribution measurements from the scanning mobility particle sizer (SMPS) indicates the ACSM sampled effectively. Masses of the refractory BC (rBC) in individual particles within the rBC-diameter range of 80 to 500 nm were measured with the Single Particle Soot Photometer (SP2), allowing the determination of size distributions of number
concentration of rBC-containing particles and mass concentrations of rBC (Sedlacek et al., 2022). CO mixing ratios were measured with the CO/NO$_2$/H$_2$O integrated cavity output spectroscopy (ICOS) analyzer. (Table 1).

Size distributions of aerosol number concentration for mobility diameters between 10 and 500 nm were measured with the SMPS, and total number concentrations of particles with diameters between 10 and 3000 nm were measured with a Condensation Particle Counter (CPC). Light-scattering coefficients ($\sigma_s$) at wavelengths 450, 550, and 700 nm were measured
with the nephelometer, and light-absorption coefficients ($\sigma_a$) at 465, 530, and 650 nm were measured with the Particle Soot Absorption Photometer (PSAP). These quantities were taken from the ARM Aerosol Optical Properties Value Added Product (AOP VAP; Flynn et al., 2020), as were values of SSA$_{530}$ that were calculated from them. The mass absorption cross-section at 530 nm (MAC$_{530}$), calculated as the ratio of the absorption coefficient at 530 nm determined by the PSAP to the rBC mass concentration determined by the SP2, is used below to quantify aerosol absorption enhancement. A mass absorption cross-
section of 7.5 m$^2$ g$^{-1}$ at 550 nm, which corresponds to ~7.8 m$^2$ g$^{-1}$ at 530 nm under the assumption of an inverse relationship between absorption coefficient and wavelength (i.e., absorption Ångström exponent (AAE) equal to unity), is universally used for fresh, uncoated rBC (Bond and Bergstrom, 2006). Therefore, a value of MAC$_{530}$ substantially greater than this indicates enhanced light absorption (Bond et al., 2013; Zanetta et al., 2016). The absorption Ångström exponent between wavelengths





470 and 660 nm ($AAE_{470-660}$) is the negative of the ratio of the logarithm of the absorption coefficient at 470 nm divided by

135  that at 660 nm to the logarithm of 470 divided by 660. These values were also taken from the ARM AOP VAP.

Table 1. Instrument and Analysis Techniques Table.

| Measurement | Instrument / Technique | Notes |
|---|---|---|
| Mass concentrations of Organic, Nitrate, Sulfate, Ammonium, Chloride particulate matter | Aerosol Chemical Speciation Monitor (ACSM) (Aerodyne Inc.) | Data only available in 2017 |
| Mass and number concentrations and coating-to-core mass ratio of black carbon-containing particles | Single Particle Soot Photometer (SP2) (Droplet Measurement Technologies) | SP2 disconnected from the sampling line after September 21, 2017. Sampled particles between 80 and 500nm |
| Size distribution and total number concentration | Scanning Mobility Particle Sizer (SMPS) (TSI Inc 3936) | Sampled particles between 10 to 500nm |
| Total particle number concentration | Condensation Particle Counter (CPC) (TSI Inc.) | Sampled particles between 10 and 3000nm |
| Aerosol light-scattering coefficient at 470, 530, and 660 nm | Nephelometer (TSI Inc.) | Data was corrected using both Virkkula and Bond & Ogren corrections. |
| Aerosol light-absorption coefficient at 450, 550, and 700 nm | Particle Soot Absorption Photometer (PSAP) (Radiance Research) | |
| $CO$, $NO_2$, mole mixing ratio | $CO/NO_2/H_2O$ Analyzer ICOS Los Gatos Research | |
| Surface wind speed and direction | Radiosonde | Launched 4x daily |
| Single Scattering Albedo at 530 nm ($SSA_{530}$) | $SSA = \dfrac{\sigma s \text{ at } \lambda 530}{(\sigma s + \sigma a)}$ | Data products available as an ARM Value Added Product (VAP) file |
| Mass Absorption Cross-section at 530 nm ($MAC_{530}$) | $MAC_{530} = \dfrac{\sigma a \text{ at } \lambda 530 \text{ Mm}-1}{black\ carbon\ (\mu gm-3)}$ | |
| $f44$, $f60$ | Extracted using positive matrix factorization and k means clustering techniques with ACSM organic aerosol data | Analysis performed at Brookhaven National Lab with ME-2 engine and SoFi software |
| Fraction of black carbon (FrBC) | $FrBC = \dfrac{rBC\ \#\ concentration}{cpc\ \#\ concentration}$ | |
| Fire count / distribution | SUOMI Visible Infrared Imaging Radiometer Suite (VIIRS) | Data available on NASA FIRMS Archive. Data binned in 2°x2° boxes |
| Meridional and Zonal winds reanalysis data | National Centers for Environmental Prediction (NCEP) | 4x Daily Average at 850mb |
| CO (ppbv) reanalysis data | Copernicus Atmosphere Monitoring Service (CAMS) | Data available on Copernicus Atmosphere Monitoring Service archive |



## 2.2 Data Analysis Techniques

$\Delta$CO mixing ratios at Ascension Island were calculated by subtracting the 5[th] percentile value of the measured CO mixing ratio from the average value for each month, following Che et al. (2022a), and then converted to mass concentrations.

These mass concentrations, together with rBC mass concentrations measured by the SP2, were used to determine mass ratios of rBC:$\Delta$CO. These mass ratios can be used as an indicator of burning conditions at the source because rBC is chemically inert (Wang 2004; Cape et al., 2012; Lund et al., 2018) and the average lifetime of CO is approximately two months (Khalil et al., 1990), much longer than typical transport times. rBC:$\Delta$CO was used to classify fires as either inefficient or efficient, with inefficient fires having values of rBC:$\Delta$CO less than 0.01 and efficient fires having values of rBC:$\Delta$CO greater than 0.01,

consistent with previous classifications (Vakkari et al., 2018; Che et al., 2022a). Surface RH fields provided by the NOAA National Center for Environmental Prediction (NCEP) reanalysis are independently used to assess the burning condition classification. Inefficient fires typically produce relatively more OA and $SO_4$ and relatively less rBC than efficient fires (Collier et al., 2016; Rickly et al., 2022). Although modified combustion efficiency (MCE) may be a better determinant of burning conditions (Collier et al., 2016; Dobracki et al., 2023), this quantity could not be calculated because $CO_2$ was not sampled at

Ascension Island during the LASIC campaign.

The locations of the fires between 5.7 ˚W-52.2 ˚E, and 3.2 ˚N-34.6 ˚S, encompass the sources of most of the BBA measured at Ascension Island. Fire locations were determined from fire distributions and counts obtained from NASA's Fire Information for Resource Management System (FIRMS) archive, which uses the Visible Infrared Imaging Radiometer Suite (VIIRS) aboard the S-NPP and NOAA 20 satellites. The vegetation types contributing to the fuel for the fires were identified

using annual land use maps from the Moderate Resolution Imaging Spectroradiometer (MODIS), combined with the NOAA Hybrid Single-Particle Lagrangian Integrated Trajectory (HYSPLIT) back trajectories initialized with Global Data Assimilation System (GDAS) meteorology inputs. Transport pathways were determined using a combination of back trajectories from HYSPLIT and CO mixing ratios from the Copernicus Atmosphere Monitoring Service (CAMS) combined with European Centre for Medium-Range Weather Forecasts (ECMWF) reanalysis (ERA5) 950, 800, and 700 hPa winds (Figs.

S1a-d).

Positive matrix factorization (PMF) was applied to the ACSM-derived compositions during BBA-laden time periods to apportion the organic mass spectra into various factors, following Aiken et al. (2008), Lanz et al. (2010), and Zhang et al. (2011). This PMF analysis yielded only two factors: oxygenated organic aerosol (OOA) and low-volatility oxygenated organic aerosol (LV-OOA). This result is not surprising, as BBA dominated the composition during the time periods during which

PMF was applied. This analysis also provided robust calculations of the ion fractions *f44* and *f60*, with *f44* indicating the presence of the $CO_2^+$ ion, a product of oxidation (Canagaratna et al., 2015), and *f60* indicating the presence of $C_2H_4O_2$, a fragment of levoglucosan, which is a known tracer for BBA (Cubison et al., 2011). Further details on the PMF analysis can be found in Appendix A.





Monthly-mean above-background $\Delta SO_4$ mass concentrations were calculated by subtracting $SO_4$ mass concentrations

averaged over times each month when rBC mass concentrations were less than 20 ng m$^{-3}$ (Table 2), from the total $SO_4$ mass concentration during BBA-laden times (Table 2). This quantity excludes background $SO_4$ produced from marine or local sources in the MBL. Background $SO_4$ concentrations were typically higher later in the biomass burning season and did not correlate with windspeed, suggesting some remaining BBA residual $SO_4$. Changes in the mass ratios of $\Delta SO_4$ to rBC ($\Delta SO_4$:rBC), OA:rBC, and *f44* can indicate changes in BBA properties from aqueous-phase chemistry and chemical processing

during transport, respectively. Oxalate, an organic acid that is a well-known tracer of aqueous-phase oxidation of OA that contributes to *f44* and can indicate that the aerosol has interacted with cloud (Sorooshian et al., 2010; Ervens et al., 2011). However, the ACSM cannot specifically distinguish oxalate from other species that contribute to *f44*.

Table 2. $\Delta SO_4$, rBC, and $\Delta CO$ from clean periods

| Start time | End time | rBC (µg m$^{-3}$) x10$^{-4}$ | $\Delta CO$ (ppb) | $SO_4$ (µg m$^{-3}$) x10$^{-1}$ |
|---|---|---|---|---|
| 6/4/17 3:01 | 6/6/17 0:26 | 2.46 | 1.13 | 2.40 |
| 6/11/17 7:19 | 6/12/17 20:24 | 5.36 | 3.91 | 2.89 |
| 6/20/17 19:18 | 6/20/17 22:27 | 6.06 | 3.96 | 4.93 |
| 7/11/17 17:12 | 7/12/17 19:32 | 3.40 | 0.44 | 2.18 |
| 7/20/17 16:21 | 7/21/17 13:17 | 9.79 | 6.17 | 1.86 |
| 7/25/17 18:34 | 7/26/17 20:52 | 6.14 | 6.24 | 7.97 |
| 7/29/17 20:57 | 7/31/17 6:03 | 5.18 | 6.55 | 1.01 |
| 8/21/17 8:32 | 8/24/17 2:10 | 1.41 | 3.65 | 8.89 |
| 9/13/17 3:18 | 9/13/17 15:27 | 3.8 | 6.53 | 9.71 |
| 9/15/17 2:57 | 9/16/17 3:16 | 8.3 | 1.67 | 1.36 |


The number fraction of rBC-containing particles (FrBC) in the BBA was calculated by dividing the number concentrations of rBC-containing particles measured by the SP2 by the total particle number concentration measured by the CPC. This value is a lower limit because the SP2 detects only particles with rBC-core diameters between 80 and 500 nm, whereas the CPC detects particles with total diameters between 10 and 3000 nm. However, there were likely very few rBC

particles with rBC-core diameters less than 80 nm or greater than 500 nm (Taylor et al., 2020; Dobracki et al., 2023). The rBC geometric peak diameter (rBC$_{gpd}$) is defined as the diameter at the maximum value of the size distribution of number concentration of rBC-containing particles in the representation $dN/d\log D$, and the rBC core mass-equivalent peak diameter (rBC$_{mpd}$) is defined as the diameter at the maximum value of the size distribution of mass concentration in the representation $dM/d\log D$. The coating-to-core mass ratio of rBC-containing particles is defined as the daily average of the ratio of the coating

mass to the rBC core mass and has an uncertainty of ~20 % (Sedlacek et al., 2022).



## 3 Results

A time series of rBC and ΔCO spanning from June through September 2017 indicates significant synoptic variability, with times when rBC mass concentrations were near 1000 ng m$^{-3}$ interspersed with times with rBC less than 20 ng m$^{-3}$ (Fig. 1, see also Pennypacker et al., 2020). Ten plume events, denoted P1 to P10, correspond to rBC mass concentrations exceeding 150 ng m$^{-3}$, the 70$^{th}$ percentile of all rBC mass concentrations. These synoptically defined events last from two days to two weeks and may contain BBA from multiple sources, each with its own combustion history.

Temporal trends in the plume rBC:ΔCO mass ratios (Fig. 2) together with maps of fire density, surface RH, land use, and fuel type (Table 3) support a classification of the 10 events into three temporal regimes, reflecting different conditions at the beginning, middle, and end of the biomass-burning period as perceived within the remote MBL.

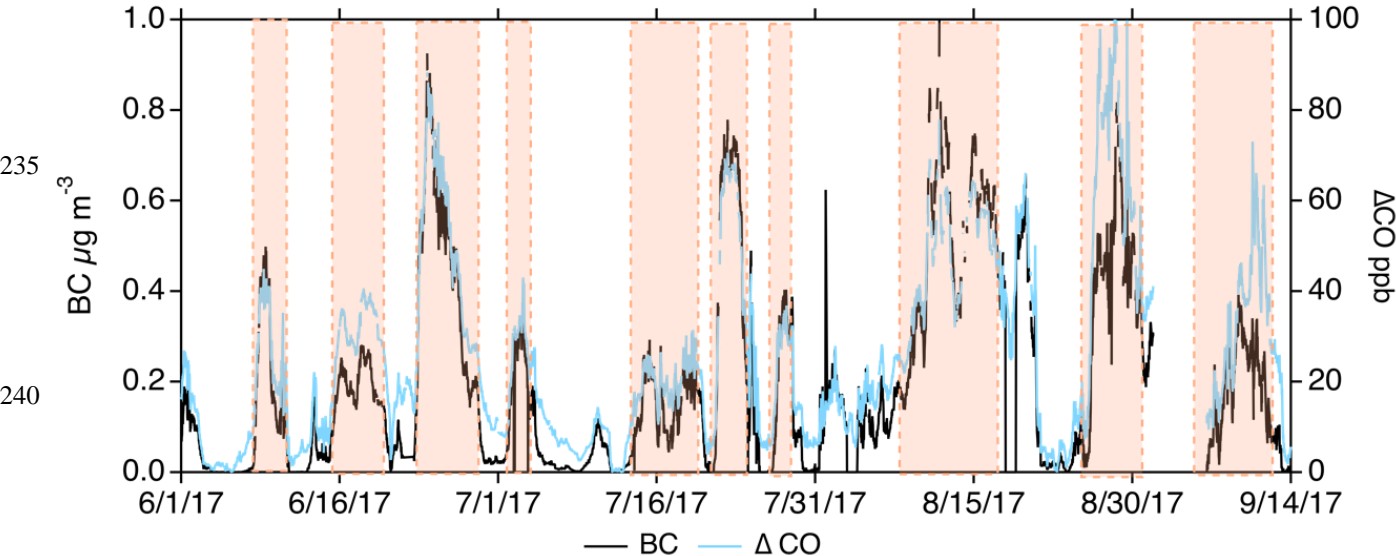

Figure 1. Time series of black carbon mass concentration (μg m$^{-3}$)(black) and ΔCO mixing ratio (ppb) (blue), calculated by removing the bottom 5th percentile of the monthly CO distribution from the total, from June 1, 2017 through September 15, 2017. Pink boxes indicate selected plume events; blue boxes indicate selected clean periods.




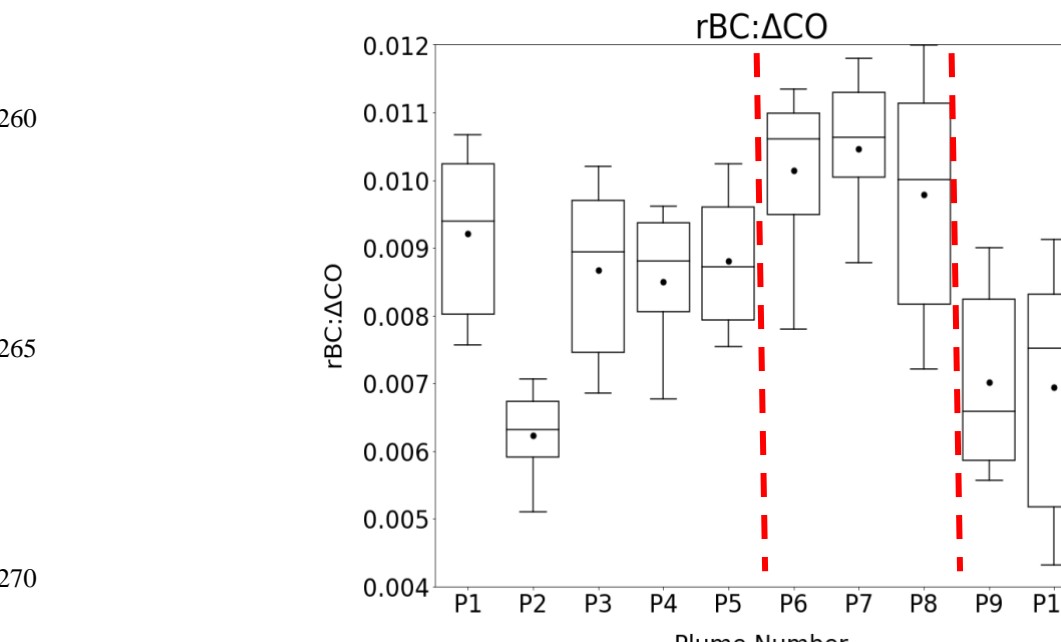



Figure 2. rBC:ΔCO box-whisker distributions for major plume events, with lowest whisker at 10th percentile, lowest bar at 25th percentile, middle bar at 50th percentile, upper bar at 75th percentile, and highest whisker at 90th percentile. Filled circles
represent the mean. Red dashed lines distinguish the three regimes discussed in the text.

Table 3. Summary of rBC, ΔCO, rBC:ΔCO, fire location, fuel type, and burning conditions for each regime description.

| | rBC (µg m$^{-3}$) | ΔCO (ppb) | rBC:ΔCO mass ratio ($10^{-3}$) | Fire location | Fuel type | Burning conditions |
|---|---|---|---|---|---|---|
| Regime 1 06/05-07/20 | 0.28±0.10 | 32±11 | 8.3±1.1 | Angola, DRC | Wood savannas and broadleaf forests | Inefficient |
| Regime 2 07/21-08/17 | 0.43±0.11 | 39±13 | 11±0.2 | Zambia, Mozambique, DRC | Grasslands and savannas | Efficient |
| Regime 3 08/24-09/11 | 0.34±0.11 | 50±20 | 7.1±0.4 | Tanzania, Mozambique, and Zimbabwe | Grasslands and savannas | Inefficient and efficient |

## 3.1 Regime-based analysis determined from rBC:ΔCO

The first five plume events, spanning June 8 to July 20, have relatively low rBC:ΔCO mass ratios of 0.0083±0.0011
(mean±standard deviation; Fig. 2), indicating inefficient combustion. Four of the events have mean rBC:ΔCO values near 0.009, with P2 significantly lower. This suggests burning conditions remained mostly homogeneous over the six weeks. Most fires were located near 10˚S and west of 30˚E (Fig. 3a; near the coast in northern Angola and in western Congo), coinciding





with surface RH values greater than 50 % (Fig. 3b). Woodier landscapes (e.g. woody savannahs and broadleaf forests) dominated in the burning region (Fig. 3c). These five events, based primarily on their similar rBC:ΔCO values, are grouped

into Regime 1.

Plume events 5-8, extending from July 21 to August 17, extend into the heart of the burning season. The mean rBC:ΔCO is higher, at 0.0112±0.0002 (Fig. 2), indicating efficient combustion. Most fires occurred south of 10˚S (Fig. 4a; in Zambia, the Democratic Republic of the Congo (DRC), and western Mozambique), coinciding with surface RH values less than 50 % (Fig. 4b). The burned regions are more grassy and less woody (Fig. 4c) than those burned in Regime 1. These plume

events are grouped into Regime 2.

The final two plume events, spanning August 24 to September 11, have the lowest overall rBC:ΔCO, with a mean of 0.0071±0.0004. Although fires are still highly active in grassland regions during this time, these values indicate inefficient combustion. Most fires occurred east of 30˚E and south of 10˚S (Fig. 5a, in northeast Zambia, southwest Tanzania, Mozambique, and Zimbabwe), with surface RH ranging between 30 and 60 % (Fig. 5b), over vegetation types varying from

grasslands to woody savannas (Fig. 5c). Most of the fires occurred over dry central Africa and many also occurred on the eastern African coast where precipitation was greater in September (Ryoo et al., 2021). These two plume events are grouped into Regime 3. A notable feature of this regime is that the strong free-tropospheric winds known as the African Easterly Jet-South became active around August 20, at approximately 700 hPa (Ryoo et al., 2022).









Figure 3. a) Regime 1 fire density maps showing 2°×2° bins of the number of fires detected by the NASA SUOMI VIIRS satellite between June 6, 2017 and July 20, 2017. b) MODIS annual (2017) mean land use map with Regime 1 fire number density contours. c) NCEP reanalysis Regime 1 mean surface relative humidity, with contours indicating fire number density.









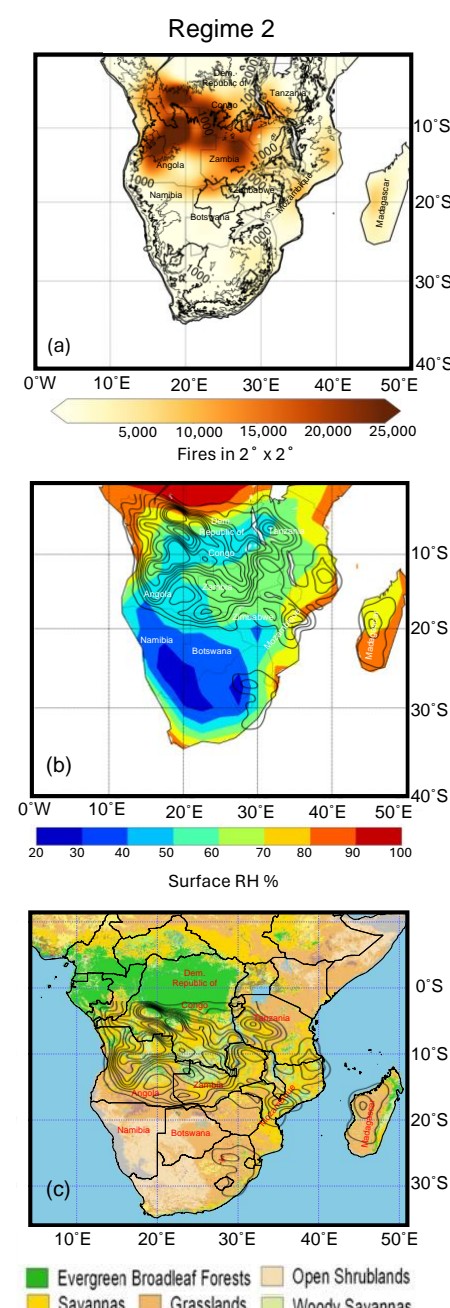

Figure 4. a) Regime 2 fire density maps showing 2°×2° bins of the number of fires detected by the NASA SUOMI VIIRS satellite between July 21, 2017 and August 17, 2017. b) MODIS annual (2017) mean land use map with Regime 2 fire number density contours. c) NCEP reanalysis Regime 2 mean surface relative humidity, with contours indicating fire number density.




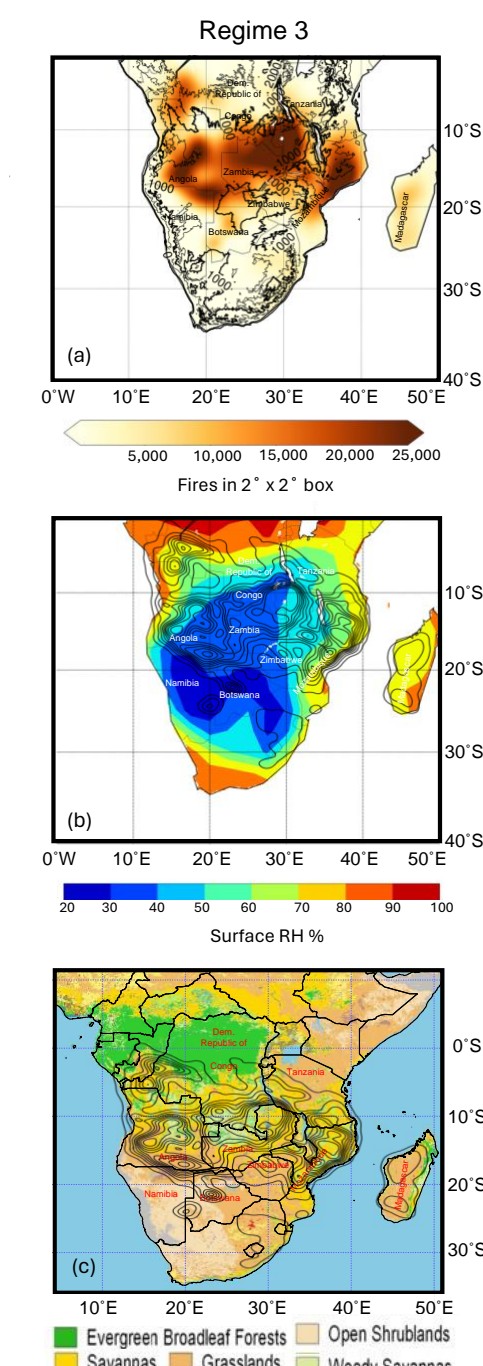

Figure 5. a) Regime 3 fire density maps showing 2˚×2˚ bins of number of fires detected by the NASA SUOMI VIIRS satellite between August26, 2017 and September 11, 2017. b) MODIS annual (2017) mean land use map with Regime 3 fire number density contours. c) NCEP reanalysis Regime 3 mean surface relative humidity, with contours indicating fire number density.



## 3.2 BBA properties measured at Ascension Island

    Here, we present the chemical, microphysical, and optical properties of the African BBA sampled in the Ascension Island MBL. By examining rBC size distributions, rBC coating-to-core mass ratios, and number fraction of rBC-containing particles, we can infer how burning conditions and fuel type varied across the three regimes (Table 4). Additionally, we
examine changes in non-refractory mass concentrations and values of *f44* across the four months to understand how long-range transport affected the chemical composition of African BBA. BBA size distributions indicate whether the BBA had undergone cloud processing or mixed with clean marine air in the MBL. Light-absorption coefficients and light-scattering coefficients are used to assess how the optical properties changed as a result of changing chemical and microphysical properties during long-range transport. Lastly, HYSPLIT back trajectories facilitate an understanding of the BBA transport pathways to
Ascension Island in June, July, August, and September.

Table 4. Summary of aerosol properties across the three regimes

|  | Regime 1 06/05-07/20 | Regime 2 07/21-08/17 | Regime 3 08/24-09/11 |
|---|---|---|---|
| $rBC:\Delta CO \times 10^{-3}$ | 8.3±1.1 | 11±0.20 | 7.1±0.40 |
| $rBC_{gmd}$ (nm) | 125±3 | 134±5 | 135±1 |
| FrBC | 0.21±0.04 | 0.29±0.04 | 0.30±0.02 |
| rBC coating-to-core mass ratio | 2.3±0.1 | 2.0±0.2 | 2.5±0.01 |
| OA:rBC | 3.3±0.73 | 2.04±0.32 | 3.5±1.8 |
| $OA:\Delta CO$ | 0.026±0.003 | 0.021±0.003 | 0.023±0.011 |
| $\Delta SO_4:rBC$ | 2.0±1.1 | 1.3±0.17 | 1.4±0.96 |
| *f44* | 0.27±0.01 | 0.26±0.01 | 0.26±0.03 |
| Acc. Dia. (nm) | 165±8 | 175±6 | 209±11 |
| $MAC_{530}\ m^2\ g^{-1}$ | 16±1 | 15±1 | 19±3 |
| $SSA_{530}$ | 0.81 0.01 | 0.78±0.02 | 0.80±0.01 |

### 3.2.1 Black carbon aerosol properties

    rBC size distributions, rBC coating-to-core mass ratios, and number fraction of rBC-containing particles provide
additional information on fuel type variations across the three regimes. The rBC geometric and mass-equivalent peak diameters $rBC_{gpd}$ and $rBC_{mpd}$, increase monotonically if undramatically from P1 to P10, with $rBC_{gpd}$ increasing from 122 to 136 nm and $rBC_{mpd}$ increasing from 193 to 203 nm, for 5-10% fractional increases from June to September (Fig. 6, Table 4). Mean FrBC also increases from 0.21±0.04 during Regime 1 to 0.29±0.03 for the latter two regimes (Fig. 7a), although internal variability is high. Interestingly, the coating-to-core mass ratio trends opposite to rBC:ΔCO, decreasing from 2.3±0.1 in Regime 1 to
2.0±0.2 in Regime 2 before increasing to 2.50±0.01 in Regime 3 (Fig. 7b). These variations suggest particles with larger rBC cores also contain a higher fraction of rBC-containing particles, with coating thicknesses that vary less with rBC core size.



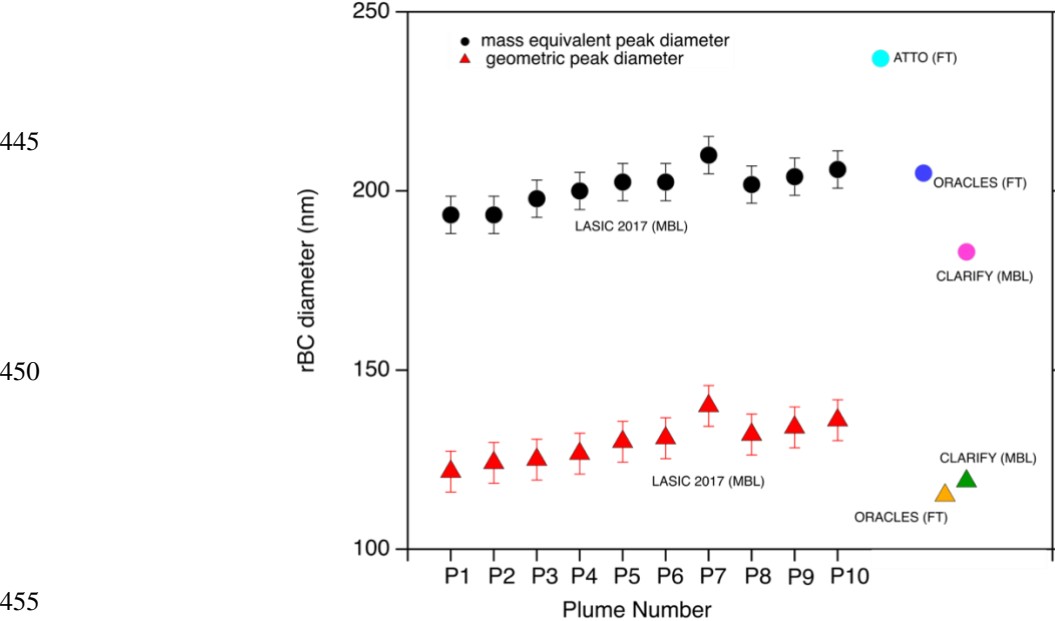

Figure 6. Mean $rBC_{mpd}$ (circles) and $rBC_{gpd}$ (triangles) for multiple field campaigns in the SEA. Filled colored circles and triangles represent the observed data from the LASIC MBL (black, red), ORACLES FT (blue, yellow) (Dobracki et al., 2023), CLARIFY MBL (pink, green) (Wu et al., 2020), and ATTO FT (cyan) (Holanda et al., 2022).

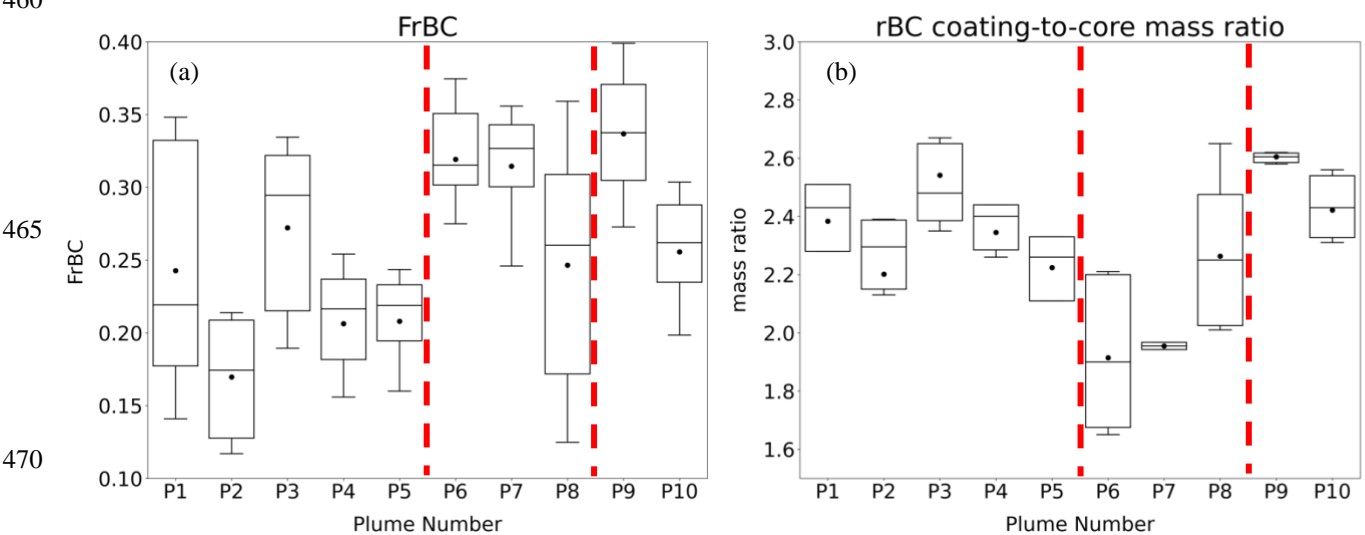

Figure 7. Box-whisker distributions of a) FrBC and b) rBC coating-to-core mass ratio with lowest whisker at 10th percentile, lowest bar at 25th percentile, middle bar at 50th percentile, upper bar at 75th percentile, and highest whisker at 90th percentile. Filled circles represent the mean. Red dashed lines distinguish the three regimes discussed in the text.





### 3.2.2 Organic and sulfate aerosol properties

OA, SO$_4$, NO$_3$, and NH$_4$ were also commonly present in the MBL at Ascension Island between June and September of 2017 (Fig. 8). Organics provided 45 % of total aerosol mass over the 10 plume events, with above-background sulfate contributing 33 %, rBC 13 %, and nitrate and ammonium the remaining 9 % (not shown). OA:rBC masss ratios generally

decreased from P1 to P9, from 3.3±1.3 (P1) to 2.2±0.8 (P9), before increasing to a maximum of 4.8±1.9, during P10 (Fig. 9a, Table 4). The trend in ΔSO$_4$:rBC approximately follows that in OA:rBC across the 10 plumes (Fig. 9b; Table 4) but is more pronounced, decreasing from 3.2±1.0 in P2 to 0.82±0.30 in P9 and then increasing to 2.2±0.9 in P10.

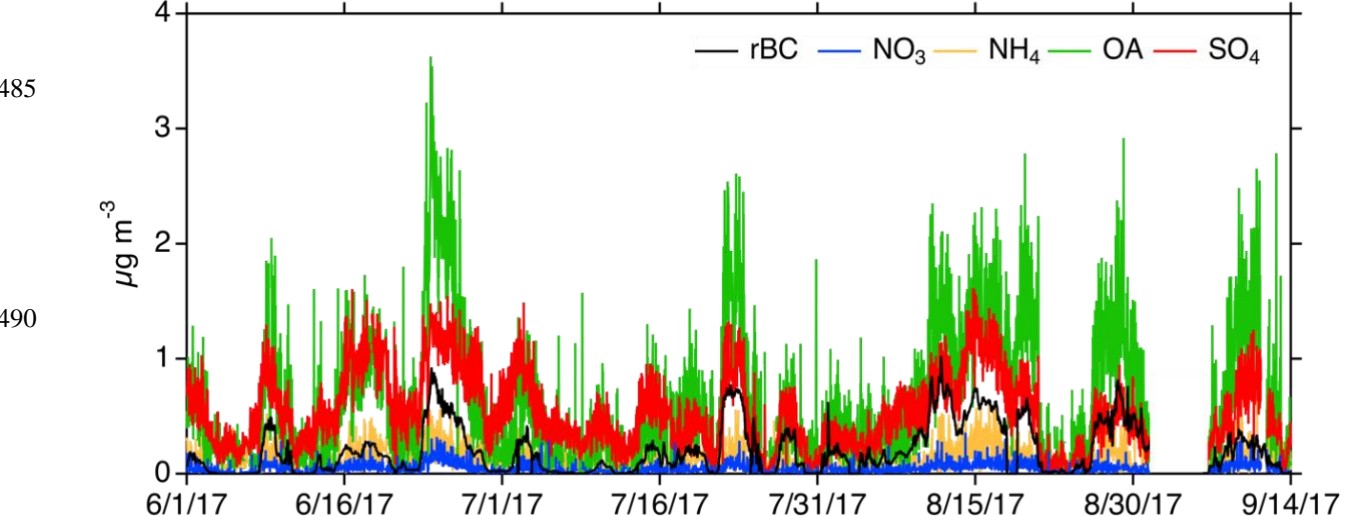

Figure 8. Time series of the mass concentrations of bulk chemical species OA, NO$_3$, SO$_4$, and NH$_4$ from the ACSM, and rBC from the SP2.

The two factors (OOA and LV-OOA) generally had similar peak intensities at *m/z* values of 18, 28, 43, and 44, and only slight differences in minor peaks (Figs. S2 and S3). Most of the plumes had similar *f44* values (Fig. S3), with the exception

of P5 and P10 (Fig. 10), for which Factor 1 *f44* values were 0.23±0.01 and 0.26±0.01, respectively, and Factor 2 *f44* values were 0.29±0.01 and 0.313±0.002, respectively. These differences possibly result from regime transitions and/or more complex fuel source mixtures. *f44* values from both Factor 1 and Factor 2 for the other 8 plumes were not substantially different; therefore, averages of the two factors are used to describe each of the 10 events. These values ranged between 0.24 and 0.28 (Fig. 10), indicating highly-oxidized BBA. Mean *f44* values decrease only slightly from 0.284±0.001 in Regime 1 to 0.26±0.03

in Regime 3 (Table 4) but may still provide information about transport and oxidation processes in the MBL.





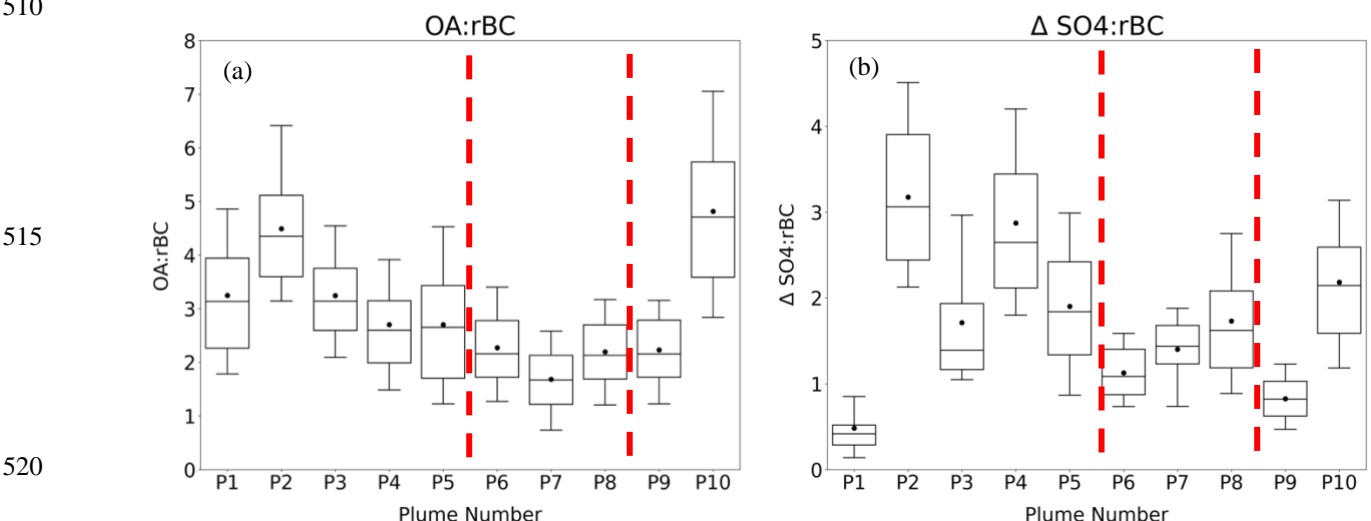

Figure 9. Box-whisker distributions of a) OA:rBC, b)$\Delta$SO$_4$:rBC with lowest whisker at 10th percentile, lowest bar at 25th percentile, middle bar at 50th percentile, upper bar at 75th percentile, and highest whisker at 90th percentile. Filled circles represent the mean. Red dashed lines distinguish the three regimes discussed in the text.

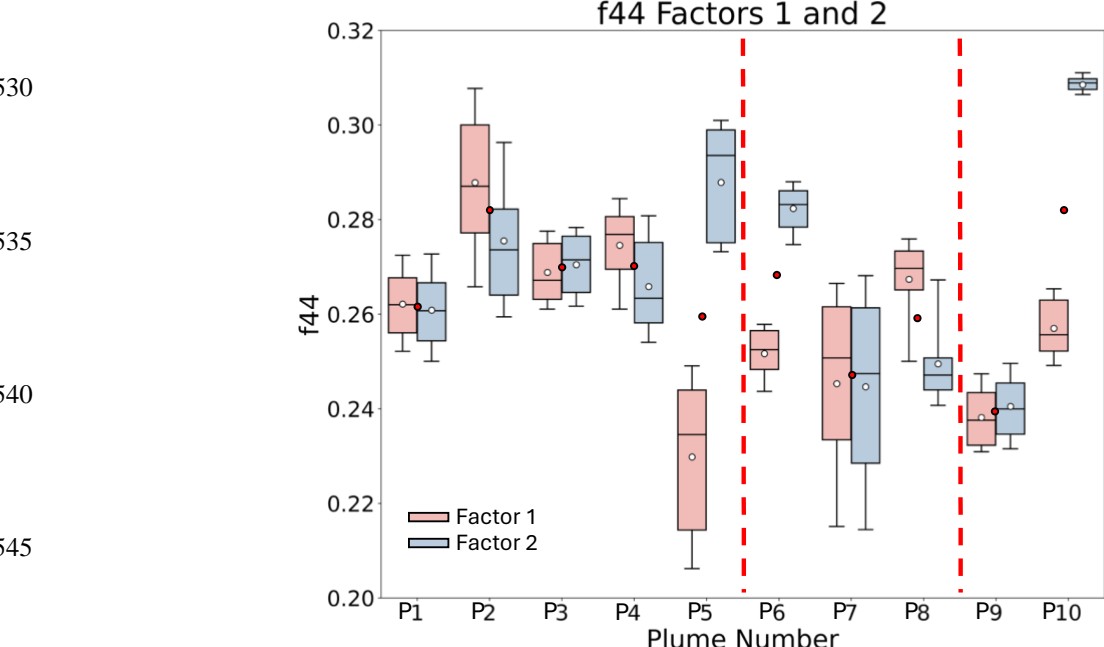

Figure 10. Box-whisker distributions of *f44* factor 1 (red) and *f44* factor 2 (blue). Filled white circles represent the mean of each factor with the lowest whisker at the 10th percentile, the lowest bar at the 25th percentile, the middle bar at the 50th percentile, the upper bar at the 75th percentile, and the highest whisker at the 90th percentile. Red dashed lines distinguish the three regimes discussed in the text. Red-filled circles represent the mean of factor 1 and factor 2.





### 3.2.3 Size distributions

The shape of the aerosol size distribution provides information on whether or not the aerosol has undergone cloud processing. A bimodal size distribution has an Aitken mode at small diameters and an accumulation mode at larger diameters, with the minimum between these modes referred to as the Hoppel minimum. The bimodality results from the activation of larger particles into cloud drops, which then either precipitate or evaporate to form even larger aerosol particles due to the uptake of other substances and coagulation after collisions with other cloud drops. The smaller particles are typically formed

from gas-to-particle conversion of organic and sulfate vapors from the ocean or the biomass burning plume, and have not activated. Therefore, bimodal aerosol size distributions in the MBL suggest aerosol that has undergone cloud processing, be it clean marine aerosol or BBA (Figs. 11, S4).

Daily-mean number size distributions during clean time periods in Regime 1 were always bimodal, with noticeable Aitken and accumulation modes (Fig. 11a). The average of all clean size distributions (in the representation $dN/d\log D$) had an

Aitken mode with a peak near 200 cm$^{-3}$ at a diameter near 35 nm and an accumulation mode with a peak also near 200 cm$^{-3}$ at a diameter near 135 nm. The Hoppel minimum diameter was near 60 nm. Daily-mean size distributions during BBA-laden times in Regime 1 were bimodal on 25 of the 28 days (Fig. S4a). The average of the bimodal size distributions during BBA-laden times had an Aitken mode with a peak 200 cm$^{-3}$ at a larger diameter, near 45 nm, and an accumulation mode with a much larger peak ranging from 400-700 cm$^{-3}$ at a diameter near 165 nm (Fig. 11a), with a Hoppel minimum diameter near 70 nm.

The pristine size distributions in Regime 2 were also bimodal (Fig. 11b) but with a more pronounced Aitken mode than during Regime 1, with number concentrations greater than 500 cm$^{-3}$, at a diameter near 35 nm. The accumulation mode peaks at slightly less than 250 cm$^{-3}$ at a slightly larger diameter, near 145 nm, but the Hoppel minimum diameter remained near 60 nm. In contrast to Regime 1, only slightly more than half (12 out of 20) of the BBA-laden days in Regime 2 had bimodal size distributions (Fig. S4b). When BBA was present, the Hoppel minimum was less pronounced than that in

Regime 1, spanning a broader range of diameters between 45 and 85 nm. The Aitken mode for the average of these bimodal size distributions was also less pronounced, with peak number concentrations less than 150 cm$^{-3}$ occurring at diameters between 30 and 50 nm. The accumulation mode peak number concentration ranged from 400 and 900 cm$^{-3}$ and occurred at slightly larger diameters than for Regime 1, near 175 nm (Fig. 11b).

Lastly, in Regime 3, the number size distributions from the clean time periods were more weakly bimodal than in

Regimes 1 and 2 (Fig. 11c), primarily because the accumulation mode was less well defined. The Hoppel minimum diameter remained between 60 and 80 nm, similar to Regimes 1 and 2. The Aitken mode for the average of these size distributions had a maximum number concentration near 250 cm$^{-3}$ at a smaller diameter, near 20 nm, than in the two other regimes, and the accumulation mode was less pronounced, with a maximum number concentration near 100 cm$^{-3}$ at diameters around 150 nm. A lower fraction of the BBA-laden days (4 out of 12) than in Regime 2 had bimodal number size distributions (Fig. S4c), and

monomodal number size distributions containing only an accumulation mode had number concentrations peaking between 400 and 800 cm$^{-3}$ at diameters near 200 nm, larger than those in Regimes 1 and 2 (Fig. 11a).





Figure 11. Plume mean SMPS aerosol size distributions (dN/dlogD cm$^{-3}$) for each selected plume event in a) Regime 1, b) Regime 2, c) Regime 3. Monthly mean distributions from clean periods are denoted as black dashed lines.





### 3.2.4 Optical Properties

There was no obvious trend in $MAC_{530}$ (Fig. 12a), $SSA_{530}$ (Fig. 12b), or $AAE_{470-660}$ (not shown) over the 10 plume events but $MAC_{530}$ and $SSA_{530}$ do correlate positively over the first 9 plume events (Fig. S5). Plume-mean $MAC_{530}$ values remained between 15 and 18 $m^2 g^{-1}$ during P1 to P9 and increased only to 20±4 $m^2 g^{-1}$ in P10 (Fig. 12a). The lowest $MAC_{530}$

value, 15±2 $m^2 g^{-1}$, was observed during P1 (Fig. 12a), but this is statistically indistinguishable from the campaign mean of 16±2 $m^2 g^{-1}$. $SSA_{530}$ varied between 0.75±0.01 and 0.83±0.02 for individual plume events, with a campaign mean of 0.80±0.01 (Fig. 12b). Lower $SSA_{530}$ values were typically associated with higher FrBC and lower OA:rBC and $\Delta SO_4$:rBC. For example, the lowest $SSA_{530}$, 0.75±0.01, in P6, also had the highest FrBC (0.32±0.04) and low OA:rBC and $\Delta SO_4$:rBC (2.3±0.8 and 1.1±0.4, respectively), whereas the highest $SSA_{530}$, 0.83±0.02, in P2, corresponded to the lowest FrBC (0.17±0.04) and high

OA:rBC and $\Delta SO_4$:rBC (4.5±1.3 and 3.2±1.0, respectively). $AAE_{470-660}$ had a campaign mean of 1.03±0.04 and varied little between plume events.

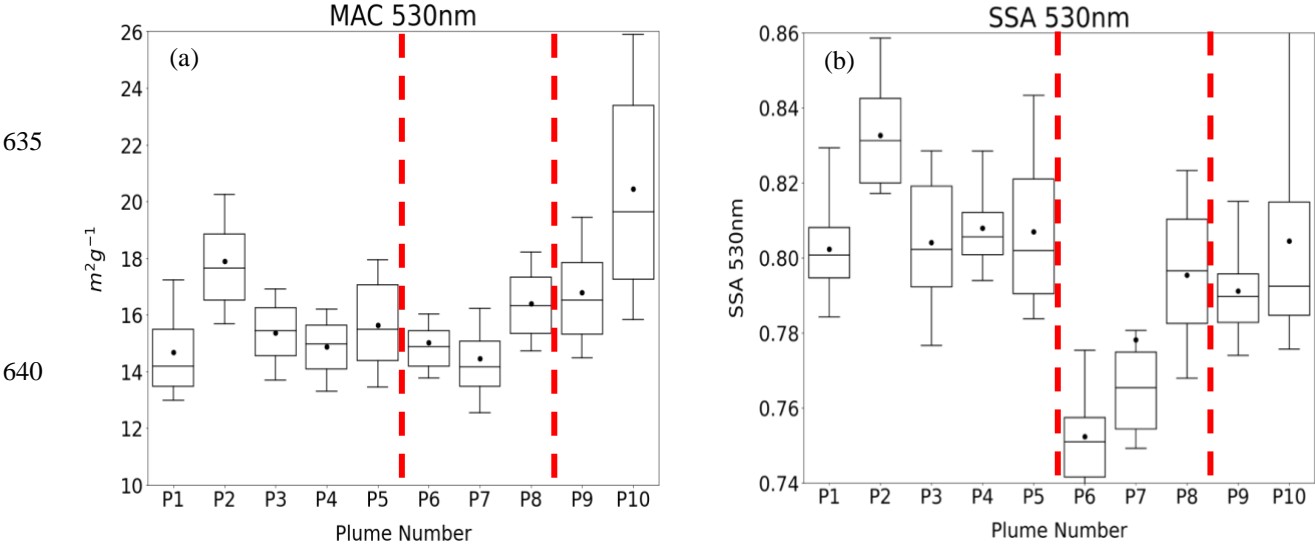



Figure 12. Box-whisker distributions of a) $SSA_{53}$ and b) $MAC_{530}$ with lowest whisker at 10th percentile, lowest bar at 25th percentile, middle bar at 50th percentile, upper bar at 75th percentile, and highest whisker at 90th percentile. Filled circles represent the mean. Red dashed lines distinguish the three regimes discussed in the text.

### 3.3 BBA transport to Ascension Island

Differences in the transport patterns between clean and BBA-laden days highlight the conditions that allow BBA to

reach the remote island. HYSPLIT back trajectories, initialized at 800, 1100, and 1400 m, are shown with NCEP reanalysis 850 hPa wind speeds for representative days with and without BBA at Ascension Island (Figs. 13a-d, Figs. 14a-d). The 8 selected days are representative of the overall transport processes occurring during June through September. Further details on the transport for each plume event using daily CAMS CO and ERA-5 winds at 900, 800, and 700 hPa are shown in Table S1.



A common feature on clean days (Figs. 14a-d) was the presence of a sea-level pressure high between 0° and 20°W,

which promoted the advection of pristine air from the southern oceans (also noted by Pennypacker et al., 2020) and forced the

BBA to north of Ascension Island. On polluted days, the lack of a sea-level pressure high allowed easterly winds to transport

BBA to Ascension Island in the upper levels of the MBL or lower levels of the FT.

Figure 13. HYSPLIT back trajectories at 800 m (red), 1100 m (blue), and 1400 m (green) above sea level for polluted days
initiated at 1200 UTC for a) June 24, 2017, b) July 23, 2017, c) August 06, 2017, and d) September 11, 2017. Each filled
colored circle represents the trajectory location at 00 UTC after initiation. Grey solid lines represented NCEP reanalysis
685 geopotential height at 850 hPa. Wind direction and magnitude are represented by arrows colored by windspeed (0-25m s$^{-1}$).
Fire density is shown in red contours for total fires two weeks prior to the initiation date.



690


Figure 14. HYSPLIT back trajectories at 800 m (red), 1100 m (blue), and 1400 m (green) above sea level for clean days initiated at 1200 UTC for clean days in e) June 5, 2017, f) July 26, 2017, g) August 22, 2017, and h) September 14, 2017. Each filled colored circle represents the trajectory location at 00 UTC after initiation. Grey solid lines represented NCEP reanalysis geopotential height at 850 hPa. Wind direction and magnitude are represented by arrows colored by windspeed (0-25m s⁻¹).
Fire density is shown in red contours for total fires two weeks prior to the initiation date.

In June, the BBA transport to Ascension Island occurred mostly near 850 hPa, in the upper levels of the MBL, with a transit time of approximately 10 days. The lack of a southeast Atlantic sea-level pressure high near the African continent allows a direct if slow westward movement of BBA. Back trajectories from June 24 (Fig. 13a) demonstrate the BBA originated





from northern Angola and the southern region of the DRC, in agreement with the conclusion above based on fire density maps (Fig. 3a).

The BBA transport in late July also occurred near 850 hPa, but because the MBL was shallower than in June (Zhang and Zuidema 2021), more of the transport may have occurred in the lower FT, with the BBA later becoming entrained into the MBL further away from the continent. The transport took less time (~8 days) because of stronger winds further south near

20˚S (Fig. 13b), despite most of the fires being located more inland, in central Africa near southern Angola and from the central region of the DRC, similar to the monthly-mean fire density maps (Fig. 4a). The BBA transport in early August was similar, also occurring near 850 hPa and requiring ~8 days. However, the fire locations shifted further south and east to southern Angola and western Zambia (Fig. 13c), also seen on the fire density maps (Fig. 5a).

The African Easterly Jet South became active after Aug. 20 (Ryoo et al., 2022) causing a dramatic switch in the BBA

transport to much higher in the FT, near 700 hPa (Ryoo et al., 2022). This would imply that little BBA was transported to the MBL at Ascension Island, despite more African fires occurring in September than any other month (Fig. 5a). Nevertheless, one last BBA-laden episode was observed in early September (Fig. 12d). The back trajectories on September 11 suggest that the BBA may have arrived after ~8 days from fires originating in Mozambique and the DRC. This last plume event is further investigated in Sect. 5.1.

**4. Discussion**

The continuous sampling of BBA in the MBL during the LASIC campaign provides a unique opportunity to characterize the evolution of the African BBA properties across the biomass-burning season. We first describe how burning conditions and fuel type, as inferred from rBC:$\Delta$CO (Sect. 3.1), affect the microphysical and chemical properties of BBA. We then discuss how heterogeneous oxidation and evaporation, along with transport pathways and aqueous-phase processes, alter

the composition and masses of the BBA particles by the time they reach the Ascension Island MBL. Lastly, we examine the dependence of $SSA_{530}$ on the chemical properties of the BBA and explain why the BBA in the MBL has a lower $SSA_{530}$ than that in the FT. Throughout this discussion, we also compare our results to those from the ORACLES and CLARIFY campaigns, which sampled African BBA in the FT and MBL between the African coast and Ascension Island, and those from the Amazonian Tall Tower Observatory (ATTO) site (Holanda et al., 2020, 2023), which sampled African BBA in Brazil, beyond

Ascension Island, in September 2014. These comparisons are summarized in Table 5 and are discussed further below.






Table 5. Field campaign comparison table

| | LASIC 2017 | | | | CLARIFY (August 2017 MBL) | ATTO (LPL) |
|---|---|---|---|---|---|---|
| | LASIC (June 2017) | LASIC (July 2017) | LASIC (August 2017) | LASIC (September 2017) | | |
| Age (days) | 10+ | 8-10 | ~8 | ~8-10 | -- | ~10 |
| $f44$ | 0.27±0.01 | 0.26±0.02 | 0.25±0.01 | 0.29±0.005 | 0.2-0.25 | ~0.30 |
| OA (mass frac. %) | 43 | 39 | 45 | 50 | 52 | 51 |
| $SO_4$ (mass frac. %) | 37 | 38 | 32 | 30 | 26 | 23 |
| $NO_3$ (mass frac. %) | 3 | 3 | 2 | 4 | 2 | 1 |
| $NH_4$ (mass frac. %) | 7 | 6 | 6 | 5 | 9 | 13 |
| rBC (mass frac. %) | 10 | 15 | 14 | 11 | 11 | 11 |
| OA:rBC | 3.34±0.7 | 2.27±0.4 | 2.32±0.2 | 4.82±1.85 | 3.8-6.25 | 4.7±0.7 |
| rBC:ΔCO (x10$^{-3}$) | 8.7±2 | 9.7±0.7 | 8.7±3 | 7±2 | 6-11 | 3±2 |
| $SSA_{530}$ | 0.81±0.02 | 0.79±0.03 | 0.79±0.02 | 0.80±0.03 | 0.8-0.86(658nm) | -- |
| $MAC_{530}$ | 16±1.7 | 15±0.5 | 16.6±1.8 | 20.45±4.3 | -- | -- |
| Median dia. (nm) | 167±9 | 167±7 | 191±15 | 217±15 | 197-213 | 105 (optical) |
| FrBC (%) | 23±5 | 26±6 | 29±6 | 26±5 | 33±10 | 9±2 |
| $rBC_{gmd}$ (nm) | 124±2 | 132±6 | 133±2 | 136±3 | 117 | -- |



## 4.1 Dependence of BBA properties on burning conditions and fuel type

In this section, we discuss how burning conditions and fuel types change across the three temporal regimes, and how these changes affect BBA properties such as rBC size distributions, FrBC, rBC coating-to-core mass ratios, and OA:rBC and SO$_4$:rBC mass ratios. These analyses offer further BBA characterization than what has typically been presented for biomass-
burning events.

The mean rBC:ΔCO values in Regime 1 were less than 0.01, indicating that the fires across the woodlands and savannas of Angola and the southern DRC were inefficient, as concluded above (Figs. 3a, 13a). The high OA:rBC values in this regime are also consistent with this classification. The BBA in this regime also had large rBC coating-to-core mass ratios, low FrBC, small values of rBC$_{gpd}$ and rBC$_{mpd}$, and high values of SO$_4$:rBC (Sects. 3.2-3.3; Figs. 6-7). These results are
consistent with those from the Southern African Regional Science Initiative 2000 campaign (SAFARI 2000; Sinha et al., 2004), which demonstrated that woodland fires across central Africa are inefficient, emitting approximately three times as much OA mass as rBC mass.

The mean rBC:ΔCO values in Regime 2 were greater than 0.01, indicating that fires across the grasslands of Anglona, the DRC, and Zambia were efficient, as concluded above (Figs. 4a, 13b). The low OA:rBC values in this regime are also
consistent with this classification. In contrast to BBA properties from Regime 1, the BBA from efficient fires in this regime had small coating-to-core mass ratios, high FrBC, large values of rBC$_{gpd}$ and rBC$_{mpd}$, and low SO$_4$:rBC values (Sects. 3.2-3.3; Figs. 6-7). These results are also consistent with those from the SAFARI 2000 campaign, which showed that burning conditions became more efficient as the soil moisture content across Zambia decreased by 80 % between June and August and rBC emissions increased relative to those of OA (Hoffa et al., 1999; Korontzi et al., 2003).

The mean rBC:ΔCO values in Regime 3 were less than 0.01, indicating that the fires across the DRC and Mozambique were inefficient, as concluded above (Figs. 5a, 13c-d). However, the low OA:rBC values, high FrBC, large values of rBC$_{gpd}$ and rBC$_{mpd}$, and low SO$_4$:rBC values observed during P9 (late August) suggest that BBA during this plume event originated from efficient fires. A similar discrepancy is seen in P10 (early September), where high OA:rBC, large rBC coating-to-core mass ratios, and high SO$_4$:rBC would suggest that the fires were inefficient, while the large values of rBC$_{gpd}$ and rBC$_{mpd}$ and high FrBC would indicate that the fires were efficient. These conflicting results imply that despite the overall low rBC:ΔCO values in Regime 3, the BBA from earlier in this regime resulted from efficient fires across central African grasslands, whereas the BBA from later in this regime resulted from both efficient fires across the grasslands and inefficient fires near the eastern coast (Jiang et al., 2020). These intriguing BBA properties observed in early September are further investigated in Sect. 5.1. Overall, the efficient fires in late August and the combination of efficient and inefficient fires in September are consistent with
observations reported by Che et al. (2022a), who concluded that burning conditions in this region become less efficient as cloud cover increases, precipitation increases, surface wind speed decreases, and soil moisture increases from August to October.



## 4.2 Black carbon core size properties

Average $rBC_{mpd}$ values from the current study generally compare well with those from other studies in this region (Fig. 6). The average $rBC_{mpd}$ value of 180 nm observed in the MBL in August during the CLARIFY campaign is slightly smaller (although still within instrument uncertainty) than the corresponding LASIC value of 200 nm, while the average $rBC_{mpd}$ value of 200 nm observed in the FT closer to the African coast in September during the ORACLES campaign (Dobracki et al., 2023) is the same as that sampled during LASIC at this time. The average $rBC_{mpd}$ value of 238 nm sampled at the ATTO site,

further west, in September of 2014, was considerably larger than that sampled between the African coast and Ascension Island during LASIC, CLARIFY, and ORACLES (Fig. 6). Explanations include differences in burning conditions and fuel types, which seems less likely than a combination of burning and meteorological conditions that select for larger rBC (Adebiyi and Zuidema, 2016), possibly enhanced by opportunities for the coagulation of rBC cores over the ~ 10 day transport time.

        Both the $rBC_{gpd}$ and $rBC_{mpd}$ increased between June and September, with the largest values of 140 and 210 nm,

respectively, occurring when the rBC:ΔCO mass ratio was the highest (0.011) in P7 (late July). This suggests that efficient fires produce either larger rBC core diameters or more and smaller rBC particles that coagulate to form larger rBC particles as the BBA undergoes long-range transport (Pan et al., 2017). However, large $rBC_{gpd}$ and $rBC_{mpd}$ values of 135 and 205 nm, respectively, were also observed during Regime 3, in late August and early September when burning conditions were less efficient (rBC:ΔCO = 0.008). These results are consistent with Holder et al. (2016), who conclude woodier, inefficient fires

across the southeast United States emit larger rBC particles than those from efficient grassland fires. This contradicts this study's findings that $rBC_{gpd}$ and $rBC_{mpd}$ values were smaller (125 and 195 nm) when fires were also less efficient (rBC:ΔCO = 0.009). These contrasting results suggest that further research is needed on the dependence of rBC core diameters on burning conditions and fuel types.

## 4.3 Heterogeneous oxidation as an aerosol mass removal mechanism in the MBL

The evolutions of *f44* and *f60* help quantify changes in aerosol chemical composition from heterogeneous and aqueous-phase oxidation, and thus to determine if heterogeneous oxidation is a dominant chemical process that can explain the low mass ratios of OA:rBC (Fig. 9a). In fresh BBA, *f44* values are typically near 0.05 and those of *f60* are typically near 0.04 (Cubison et al., 2011; Garofalo et al., 2019). Heterogeneous oxidation can increase *f44* through the formation of carboxylic acid groups, and decrease *f60* through the fragmentation of levoglucosan ($C_6H_{10}O_5$) (Ng et al., 2010; Canonaco et al., 2015).

Heterogeneous oxidation can also fragment carbon-carbon bonds in the OA, allowing the smaller fragments to subsequently evaporate, because of the higher vapor pressure of compounds with lower molecular weight (Kroll et al., 2009; 2015).

        Aircraft data from the Arctic show that fragmentation processes can increase *f44* from 0.05 to 0.12 and decrease *f60* from 0.04 to 0.02 within 5 hours (Fig. 15; Cubison et al., 2011). ORACLES data show *f44* increasing from 0.18 to 0.22 and *f60* decreasing from 0.006 to 0.004 after 4 to 6 days of transport in the free troposphere (Fig. 15; Dobracki et al., 2023). This

study also shows *f44* increasing and *f60* decreasing with increasing transport time. Eight-day-old plumes at Ascension Island



had *f44* values near 0.24 and *f60* values near 0.005, whereas *f44* values for ten-day-old plumes, occurring in June, were near 0.29, and *f60* values were lower, near 0.003 (Fig. 15). The coupled temporal increase in *f44* and decrease in *f60* with increasing transport time observed in both the ORACLES and the LASIC campaigns demonstrate that heterogeneous oxidation may still be continuing for up to ten days, with contributions from aqueous-phase oxidation discussed in Section 4.4. This result is

particularly noteworthy because to date, few studies have observed evolution in *f44* for times greater than a few hours (see also Che et al., 2022b; Dang et al., 2022).

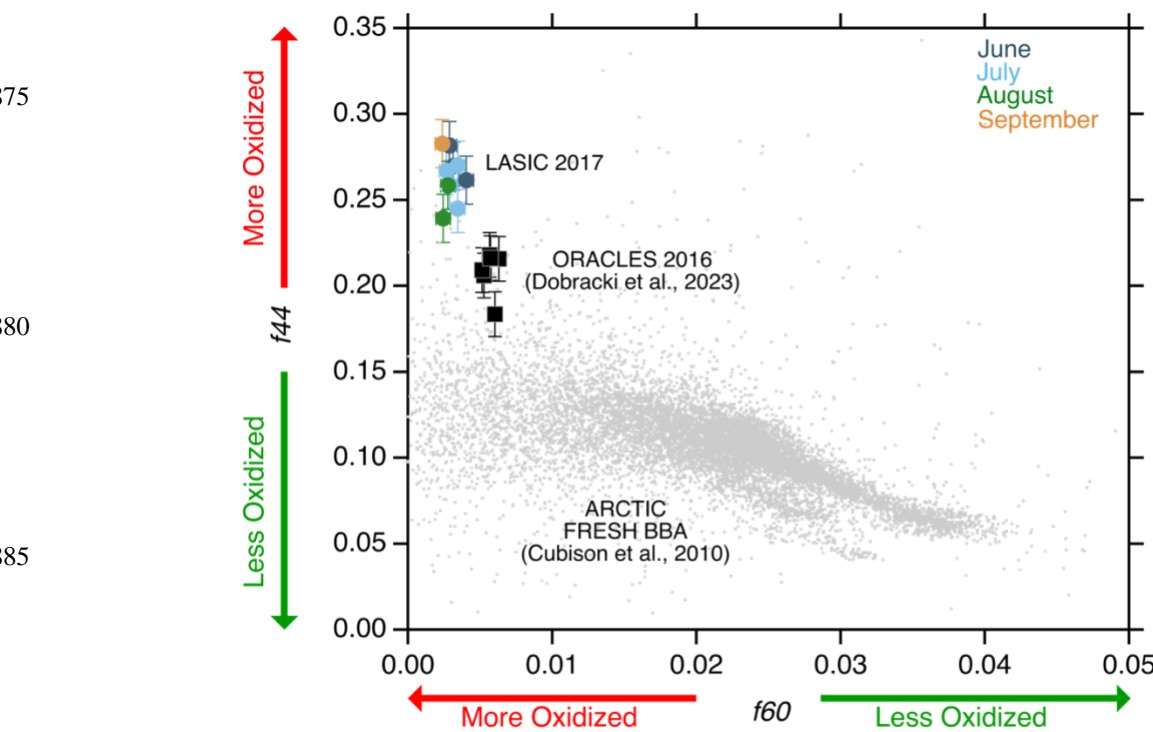

Figure 15. *f44* versus *f60* with LASIC plume event data (filled circles), colored by month, ORACLES 2016 level flight average data where OA > 20µgm$^{-3}$ (black squares) (Dobracki et al., 2023), and fresh BBA from the Arctic (grey dots) (Cubison et al., 2010). Error bars represent the standard deviation of the data set.

Changes in *f44* and *f60* indicate that oxidation has occurred, but do not capture simultaneous loss of OA with transport

time and oxidation. Fragmentation has previously been invoked to explain why OA:rBC mass ratios decrease by 30% in the FT between the African coast and the ORACLES sampling locations (Dobracki et al., 2023). Here, the relationship between changes in *f44* and in the OA:rBC mass ratio during the ORACLES, CLARIFY, and LASIC campaigns and at the ATTO site is used to determine the extent of OA mass loss caused by heterogeneous oxidation across the SEA (Fig. 16). *f44* values were typically near 0.18 and OA:rBC values were near 8 for BBA sampled near the African coast during the ORACLES campaign

(Dobracki et al., 2023). *f44* values were greater than 0.22 and OA:rBC values were less than 5 for BBA sampled in the FT and




MBL near Ascension Island (Wu et al., 2020; Dang et al., 2022). *f44* values increase further to near 0.30 while OA:rBC values decrease to less than 2.5 for BBA sampled at the ATTO site in Brazil (Holanda et al., 2020). As removal through precipitation does not occur in the southeast Atlantic free troposphere during the June-September time period (Adebiyi et al., 2015), these results suggest that heterogeneous oxidation is likely the primary process causing loss of OA across the SEA and that it

continues to occur at time scales longer than four days. Loss of OA mass, also attributed to heterogeneous oxidation, is also evident in decreasing rBC coating-to-core mass ratios with increasing transport time (Sedlacek et al., 2022).



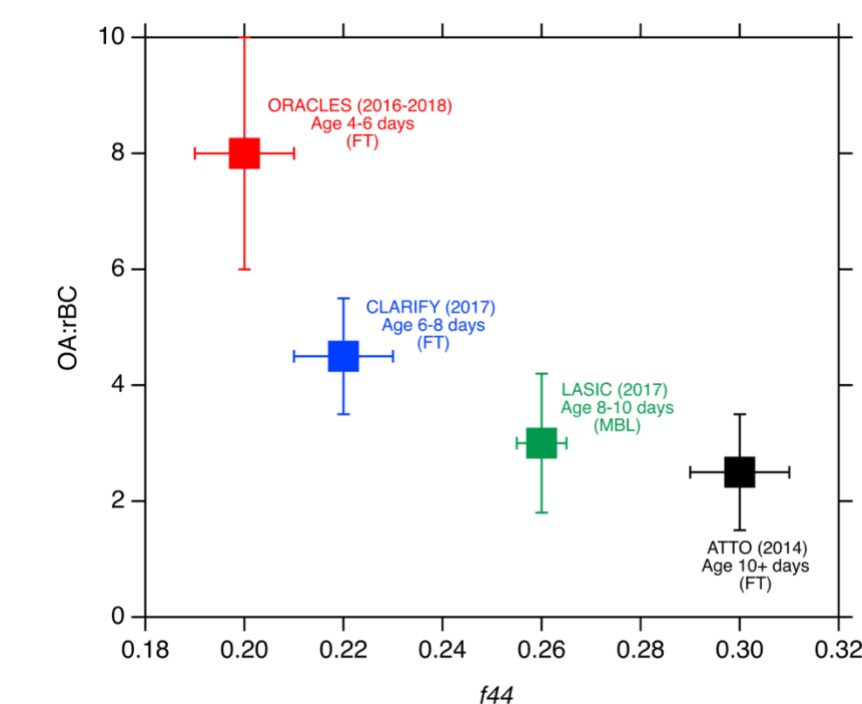



Figure 16. OA:rBC versus *f44* for the FT during ORACLES (2016-2018)(red), CLARIFY (2017) (blue) (Wu et al., 2020), and ATTO (2014)(black) (Holanda et al., 2020), and for the LASIC MBL (2017) (green dash). Error bars represent the standard deviation of the data set.


The rBC coating-to-core mass ratios between 2 and 4 observed during LASIC were also much lower than the values between 5 and 10 observed in the FT for African BBA aged between 7 and 10 days (Sedlacek et al., 2022). This indicates that most of the coating had been removed by the time the BBA reaches Ascension Island.

One other aerosol removal mechanism we consider in the MBL is through thermodynamic evaporation. Results from

the ORACLES campaign found OA:rBC decreases from only 14.5±2.6 to 11.7±1.9 as the air temperature increased from 5 ˚C to 35 ˚C (Fig. S6), suggesting some thermodynamic evaporation may contribute to OA mass loss. Given MBL air temperatures remain between 20 and 25 ˚C, evaporation is likely not an important loss mechanism, however. Another reason why the OA:rBC ratio is substantially lower in the MBL than in the FT may be higher relative humidities in the MBL encourage faster heterogeneous oxidation (fragmentation) (Wong et al., 2015; Li et al., 2018). Ascension Island RH regularly exceeds 65 %





(Zhang and Zuidema., 2021). During ORACLES, however, OA:rBC mass ratios are 10 % greater when RH is greater than

60 % (Dobracki et al., 2023), again suggesting humidity-induced oxidation may not be dominating. We next consider aqueous-

phase processing.

**4.4 Aqueous-phase processes and aerosol transport pathways are powerful determinants of aerosol size distributions, *f44,* and SO$_4$ variability**

BBA in the MBL have opportunity to directly interact with the semi-permanent deck of marine stratocumulus clouds

between the African coast and Ascension Island during their transit. These aerosol-cloud interactions could encourage a net

loss of BBA through wet removal (the formation of cloud droplets subsequently removed by precipitation). A previous

comparison of the BBA size distributions in the FT and in the MBL near Ascension Island does not indicate a preferential

removal of larger particles, as expected during precipitation events (Wu et al., 2020). Instead, non-precipitating aerosol-cloud

interactions can alter both the chemical and the physical properties of the BBA. For example, oxidants such as OH, commonly

found in high concentrations in cloud drops, can increase *f44* (McNeill et al., 2015) and alter aerosol composition. Additionally,

OA and SO$_4$ and their gaseous precursors can interact with H$_2$O$_2$ within cloud drops to form additional aerosol mass (Yang et

al., 2011; Bianco et al., 2020). Here, we use the aerosol size distributions, transport pathways, and values of *f44* to explore the

extent to which aqueous-phase processes may have occurred and contributed to additional OA loss, also helping to explain the

low OA:rBC values.

The BBA size distributions observed in the MBL were monomodal, weakly bimodal, or distinctly bimodal, whereas

the BBA size distributions observed in the FT were uniformly monomodal and broad, with most of the particles having

diameters greater than 100 nm but with a small fraction of particles having diameters less than 60 nm (Kacarab et al., 2019;

Howell et al., 2021; Dobracki et al., 2023). The monomodal BBA size distributions in the MBL indicate that this BBA likely

had not interacted with marine clouds. The weakly bimodal BBA size distributions with indistinct Hoppel minima in the MBL

(Fig. 11) indicate that this BBA had undergone insufficient processing to fully alter the shape of the distinctly bimodal

background aerosol size distributions (Atwood et al., 2020). Lastly, the distinctly bimodal BBA size distributions with

pronounced Hoppel minima (Figs. 11, S4) suggest that this BBA had undergone extensive interactions with clouds.

Regime 1 was dominated by bimodal aerosol size distributions, which, along with inferred transport pathways, high

*f44* values, and high SO$_4$:rBC mass ratios, suggest BBA that had undergone appreciable cloud processing. The BBA transport

was the slowest of the three regimes, and because upper-level FT winds were still weak, likely confined the BBA to the deeper

MBL (Zhang and Zuidema 2021). These conditions allow more time for interaction with clouds. The highest fraction of

activated aerosol particles occurred during this time (Zuidema et al., 2018), consistent larger accumulation-mode aerosol

(Dedrick et al. 2024). Relatively high *f44* values (Fig. 10), suggest some oxidation also occurred through aqueous-phase

processing (Giorio et al., 2017; Che et al., 2022b). The SO$_4$:rBC mass ratios were also the highest during Regime 1, consistent

with more conversion of SO$_2$ gas (co-emitted with the BBA) into SO$_4$ aerosol within cloud drops (Yang et al., 2011; Bianco





et al., 2020). These observations support the conclusion that the BBA in this regime likely experienced cloud oxidation and that additional aqueous-phase aerosol production.

Thereafter, fewer and less distinct bimodal aerosol size distributions, faster transport pathways, lower *f44* values, and
lower SO$_4$:rBC mass ratios in Regime 2 indicate that the BBA has undergone less cloud processing than in Regime 1. The MBL is also slightly shallower (Zhang and Zuidema, 2021), allowing the BBA to remain in the FT for longer. These conditions discourage opportunities for aerosol-cloud microphysical interactions, despite a cloud fraction near Ascension Island (Zhang and Zuidema., 2021). Values of *f44* and SO$_4$:rBC in Regime 2 were lower than those in Regime 1, also indicating less additional oxidation through aqueous-phase processing.

During Regime 3, mostly monomodal aerosol size distributions, faster transport pathways in the FT, lower *f44* values, and lower $\Delta$SO$_4$:rBC mass ratios suggest that the BBA has not undergone extensive cloud processing. BBA is more likely to be transported above the MBL, in part by stronger FT easterly winds after August 20, and thereby less likely entrained into the MBL until near Ascension Island (Ryoo et al., 2022). Delayed entrainment into the MBL would result in the BBA replacing the marine aerosol initially, with little mixing.

Cloud processing, which is most active in Regime 1, can also produce small amounts of additional aerosol mass, increasing the mass fractions of OA and $\Delta$SO$_4$ (Ervens et al., 2011; Gilardoni et al., 2016). This may have occurred during Regime 1, for which rBC particles also possess a slightly greater coating-to-core mass ratios than in the latter regimes, and for which OA:rBC and *f44* correlate positively ($r^2 = 0.61$), as do $\Delta$SO$_4$:rBC and *f44* ($r^2 = 0.58$; Fig. S7). However, the correlations between OA:rBC and *f44* and between SO$_4$:rBC and *f44* could also have been influenced by differences in aerosol composition
at the fire sources and differences in transport pathways. Nevertheless, our conclusions that aqueous-phase processing in non-precipitating clouds oxidizes the OA but does not substantially remove it are consistent with the results of Che et al. (2022b) and numerous laboratory studies (Ervens et al., 2011; Gilardoni et al., 2016). The smaller values of OA:rBC in the MBL compared to the FT, in August, are most likely explained by a combination of longer transport paths allowing for more heterogeneous oxidation, including within clouds.

**4.5 SSA$_{530}$ values in the MBL are best explained by OA:rBC**

The SSA$_{530}$ values in the Ascension Island MBL, between 0.75 and 0.83 (Section 3.2.4), are among the lowest in the world from BBA (Eck et al., 2013; Zuidema et al., 2018) and will strengthen the boundary layer semi-direct effect (Zhang and Zuidema, 2019) as a result. These values are even lower than those in the nearby FT by about 0.07. This result is surprising, as mixing with clean marine air should increase SSA$_{530}$ in the MBL all else equal (Wu et al., 2020). SSA$_{530}$ is negatively
correlated with FrBC (Fig. 17a) and positively correlated with OA:rBC (Fig. 17b) in both the MBL and the FT near Ascension Island. Moreover, SSA$_{530}$ is not correlated with the geometric peak diameter of the accumulation mode (Fig. S8) in either the MBL or FT, consistent with Denjean et al. (2020b) using in-situ data over the Gulf of Guinea and Dobracki et al. (2023). FrBC values for both the MBL and the FT are almost identical between ORACLES and LASIC, ranging between 0.15 and 0.40, so variations in FrBC cannot account for the reduced MBL SSA$_{530}$ values either (Fig. 18a). Instead, one difference is that the





OA:rBC  mass ratios in the MBL (ranging from 2 to 5) are markedly lower than those in the FT (between 5-15). Of further note, a robust correlation exists between $SSA_{530}$ and OA:rBC that applies to both the MBL and FT: $SSA_{530} = 0.78 + 0.0077 \times OA:rBC$ (Fig. 18b). This relation is similar to that proposed by Dobracki et al. (2023) for the ORACLES 2016 campaign. The $SSA_{530}$ dependence on OA:rBC shows that accurate estimates of $SSA_{530}$ can be obtained from the BBA chemical properties alone.

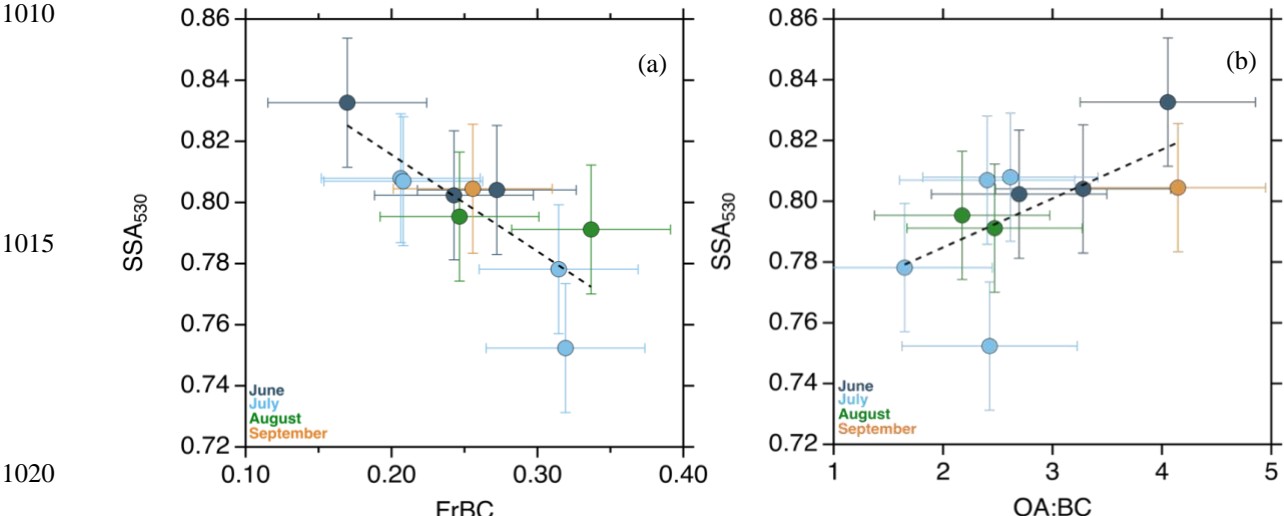

Figure. 17 a) $SSA_{530}$ versus FrBC b) $SSA_{530}$ versus OA:rBC. Markers are colored by month and error bars represent the standard deviation of the data set. The Black dashed line is the best fit line calculated by linear regression analysis. The best-fit line is weighted by µg m⁻³ min⁻¹.

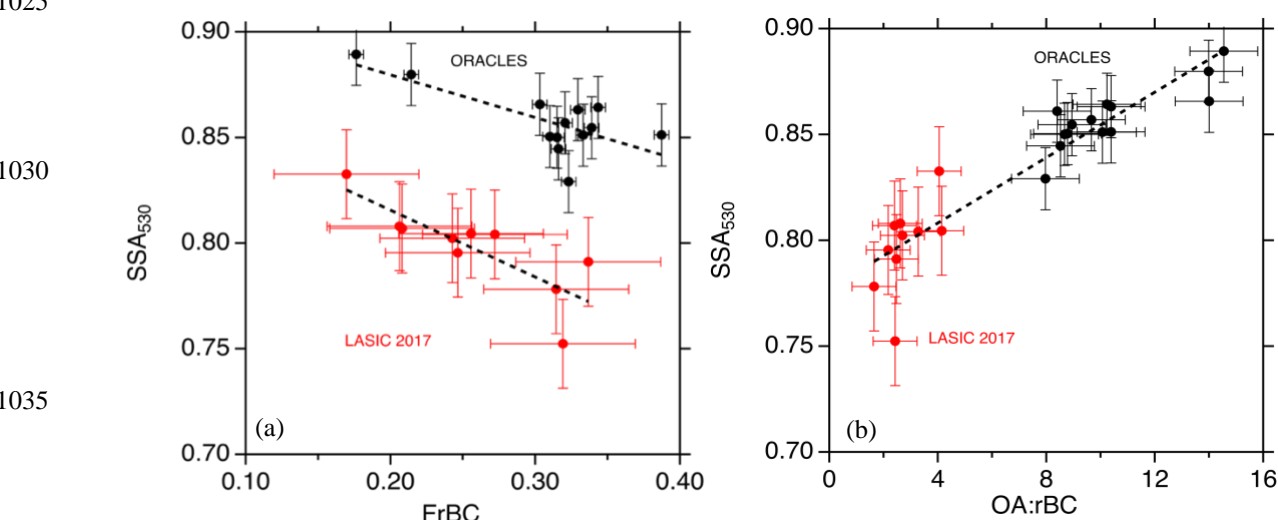

Figure 18. a) SSA 530 nm versus OA:rBC for ORACLES level leg data (black) (Dobracki et al., 2022) and LASIC 2017 major plume event data (red). b) SSA530 nm versus OA:rBC for ORACLES level leg data (black) and LASIC 2017 major plume event data (red). Error bars represent the standard deviation of the data set and the black dashed line is the best-fit line calculated by linear regression analysis.




## 5. Remaining Questions

### 5.1 Case Study: 6-11 September 2017


The last plume event to reach the MBL at Ascension Island between September 6-11 (P10), is unique in that the BBA originated from both efficient fires across central Africa and inefficient fires along the eastern coast of Africa (Section 4.1). This plume is notable for possessing large rBC core diameters (Fig. 6), and the largest OA:rBC mass ratio (Fig. 9), $f44$ values (Fig. 10; factor 2 only), and MAC$_{530}$ of the 10 plumes.


Additional HYSPLIT back trajectories (Fig. 19) indicate that the BBA sampled before September 10 may have experienced more time in the FT than that from 1-2 days later, by first circulating anticyclonically ~10° to the southeast of Ascension Island in the lower FT (1-2.5 km) before entering the MBL (Fig. 19a), similar to the case discussed in Diamond et al. (2022). The African Easterly Jet South would have advected the BBA aloft prior to entrainment into the MBL (Ryoo et al., 2022). In contrast, the BBA sampled after September 10 was transported directly westward from the continent (Fig. 19b),


potentially from as far away as eastern Africa. This direct westward transport, meteorologically unusual for September (Ryoo et al., 2022), is more likely to occur at lower altitudes (Fig. 13d), and the different source region may explain why the aerosol properties of this plume differed from those from previous times.

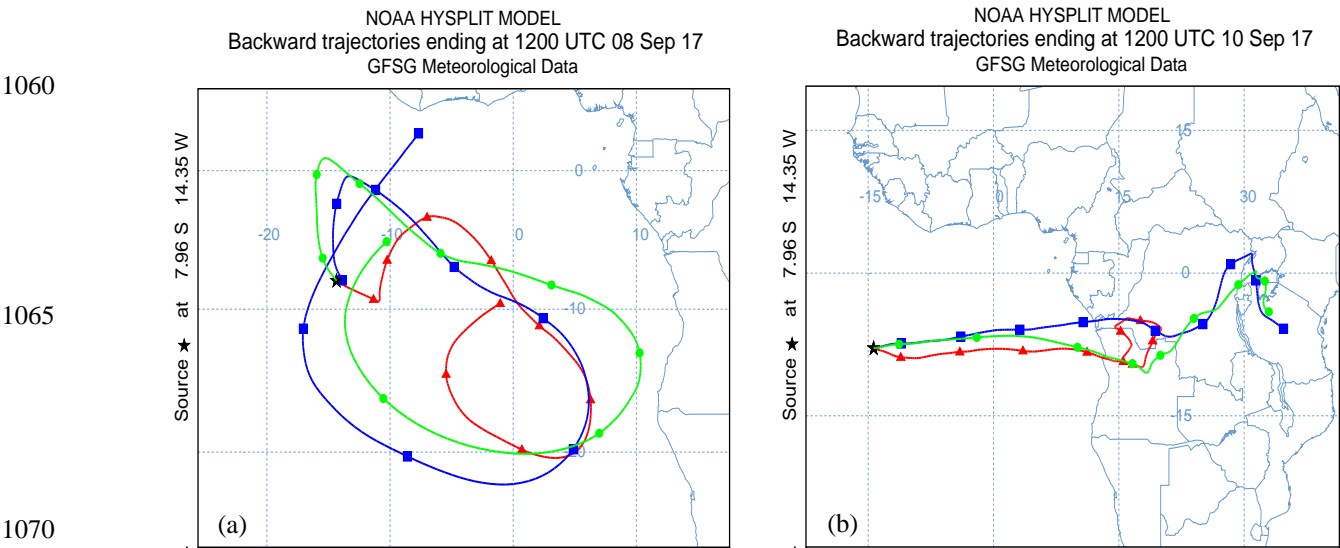

Figure 19. 10-day HYSPLIT back trajectories at 1000 m (red), 1500 m (blue), and 2500 m (green) on a) September 8, 2017 at 1200 UTC and b) September 11, 2017 at 1200 UTC.


The high values of rBC:ΔCO (Fig. S9a) and FrBC and the low values of OA:rBC (Fig. S9b) and SO$_4$:rBC (Fig. S9c) indicate that the BBA sampled before September 10 originated from efficient fires. Lower values of rBC:ΔCO and FrBC paired with the higher values of OA:rBC and SO$_4$:rBC after September 10 indicate BBA during the last two days from inefficient



combustion. Although these chemical properties are consistent with inefficient combustion, the optical properties do not.
Intuitively $MAC_{530}$ would increase with OA:rBC because of greater absorption by particles with larger OA coatings (so-called lensing) while $SSA_{530}$ would increase with OA:rBC because of relatively more scattering than absorption, for either internally or externally mixed aerosol. Instead, we find the opposite relationship. Although the $MAC_{530}$ did increase from 17.5 to 29.5 $m^2$ $g^{-1}$ between September 7 and September 10, it decreased to 25 $m^2$ $g^{-1}$ on September 11 (Fig. S9d). Meanwhile, the $SSA_{530}$ decreased from 0.835 to 0.785 across the five days (Fig. S9e). The highest $MAC_{530}$ and lowest $SSA_{530}$ occurred on the day when $\Delta CO$ was the highest (Fig. S9f). Overall $MAC_{530}$ correlates positively and $SSA_{530}$ correlates negatively with $\Delta CO$ (Fig. S10), indicating greater absorption and absorptivity by particles more deeply within the smoke plume. We cannot fully explain the underlying processes.

## 5.2 Aerosol absorption enhancement

The larger analysis of LASIC also brings the typical explanation for the high $MAC_{530}$ values ($> 15$ $m^2$ $g^{-1}$) observed in the Ascension Island MBL, of lensing-enhanced absorption from thick coatings, into question. $MAC_{530}$ does not correlate with the geometric mean diameter of the accumulation mode determined from the BBA size distributions (Fig. S11), in the presence of more constant rBC core sizes, also concluded by Denjean et al. (2020b) and Dobracki et al. (2023) for FT aerosol. The high $MAC_{530}$ values in the FT have been attributed to thick coatings (Taylor et al., 2020; Sedlacek et al., 2022), but rBC-containing particles in the MBL are more thinly coated (Sedlacek et al., 2022).

$MAC_{530}$ and $SSA_{530}$ are positively correlated (Fig. S5), indicating that both scattering and absorption are enhanced when soot particles are coated with transparent OA and $SO_4$ (Khalizov et al., 2009). In addition, both $MAC_{530}$ and $AAE_{470-660}$ in the MBL increase with increasing *f44* (Figs. 20a-b), suggesting that chemical processing, either through heterogeneous oxidation or aqueous-phase processing, can act to enhance aerosol light absorption, also shown in Zhang et al. (2018). The increased light absorption resulting from chemical changes must have been small, however, as average $AAE_{470-660}$ values are near 1 in both the MBL and the FT near Ascension Island, consistent with BrC contributing less than 10 % to the aerosol absorption (Taylor et al. 2020; Zhang et al. 2022). We conclude the processes contributing to aerosol absorption enhancement in the MBL are different than those in the FT and require additional study.





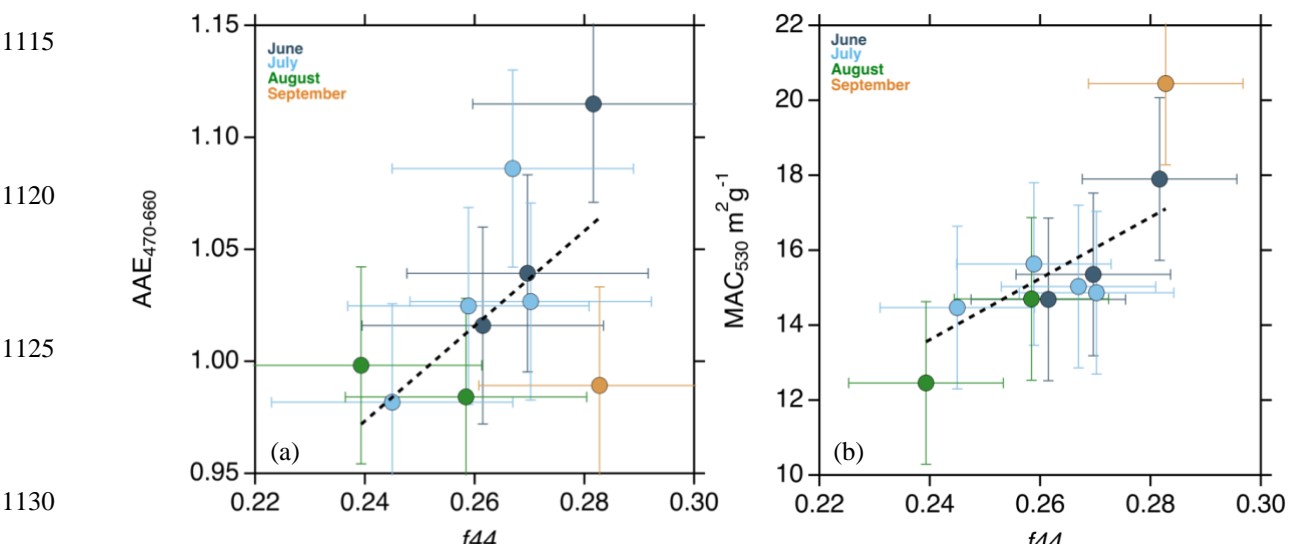

Figure 20. a) $AAE_{470-660nm}$ versus average $f44$ for the 10 selected plume events. b) Average $f44$ versus $MAC_{530}$ ($m^2$ $g^{-1}$) for the 10 selected plume events. Markers are colored by month and error bars represent the standard deviation of the data set. The black dashed line is the best-fit line calculated by linear regression analysis. The best-fit line is weighted by µg $m^{-3}$ $min^{-1}$.

## 6. Conclusions and summary

The current analysis focuses solely on 10 specific BBA plumes within the remote MBL of the southeast Atlantic, in contrast to BBA plumes in the FT. The synoptically-modulated BBA plumes in the MBL occur from June until early September, before further aerosol transport moves to the free troposphere. BBA in June and early July (Regime 1) contained more OA and $SO_4$ relative to rBC, within primarily bimodal BBA size distributions. These BBA properties resulted from inefficient burning conditions (characterized by rBC:ΔCO mass ratios less than 0.01) combined with a slow westward transport of BBA in the MBL that allowed more time for cloud processing, arguably most notable in the enhanced $ΔSO_4$ mass. The BBA in late July and early August (Regime 2) had lower contributions of OA and $SO_4$ relative to rBC, and fewer, broader bimodal size distributions, than those earlier in the season. These BBA properties resulted from efficient burning conditions (characterized by rBC:ΔCO mass ratios greater than 0.01) and from slightly faster transport pathways that were less confined to the MBL, reducing time for aqueous-phase processing. The BBA in late August and early September (Regime 3) had the highest contribution from OA relative to rBC. Mostly monomodal BBA size distributions indicate little mixing with marine air. BBA properties reflect both efficient and inefficient burning conditions. These BBA properties resulted from both efficient and inefficient burning conditions and aerosol transport pathways being mostly in the FT due to the strong easterly winds during this time that did not allow for many aerosol-cloud interactions.



The BBA transport pathways to Ascension Island in the MBL and lower FT between June and September are slow, overall, and allow not only for heterogeneous oxidation processes but also for aqueous-phase processes. Continuous heterogeneous oxidation processes for up to 10 days contribute to a net loss of OA mass, with the contribution from aqueous-phase processes (in June and July) which are difficult to distinguish from fuel type variations. Although *f44* values and BBA size distributions suggest that aqueous-phase processing occurred, further effort is required to better quantify the relative contributions of heterogeneous oxidation and aqueous-phase processing on the evolution of BBA.

SSA values measured in the Ascension Island MBL, less than 0.80, are among the lowest in the world from BBA. The low OA:rBC ratios in the MBL, a consequence of oxidation processes, likely explains these low $SSA_{530}$ values and is arguably the most striking result of this study. Current global and regional aerosol models typically do not include heterogeneous and aqueous-phase oxidation processes as major aerosol loss mechanisms, but the resulting changes in chemical composition can have substantial effects on aerosol optical properties, demonstrating the need for further effort placed on constraining the loss of OA in the MBL. A remaining puzzle is a strong positive correlation between $MAC_{530}$ and $SSA_{530}$, which is not easily explained by lensing effects.

## Appendix A

## Experimental design and Instrumentation

### 1.1 Aerosol Sampling Inlet

The inlet used in each AOS was based on the design from the Environmental Measurements Laboratory (Leifer et al., 1994; Uin et al., 2019). The inlet was positioned 10 m above the ground and drew a flow rate of 1000 L min⁻¹, with 150 L min⁻¹ distributed evenly among five sample lines (Uin et al., 2019). The flow through the sample lines was maintained by bypass flow with support from a vacuum source and from individual instrument flows on that line (Uin et al., 2019). The bypass flows were measured and monitored by rotameters. Trace gas measurements were sampled through a separate ½-inch tube that ran parallel to the main inlet (Uin et al., 2019, Figure 4). A comprehensive description of the AOS set up and operation within the ARM program is given by Uin et al. (2019).

### 1.2 Aerosol Chemical Speciation Monitor (ACSM)

The aerosol chemical speciation monitor (ACSM, Aerodyne Inc.) measures non-refractory particle mass concentrations of organic, nitrate, sulfate, ammonium, and chloride species in real time for particles with aerodynamic diameters between 75 and 650 nm at 30 min time resolution (ACSM handbook, Watson, 2014). The ACSM has a detection limit of 0.3 µg m⁻³ (30-minute signal average) for organic aerosol, 0.2 µg m⁻³ for nitrate, 0.4 µg m⁻³ for sulfate, 0.5 µg m⁻³ for ammonium, and 0.2 µg m⁻³ for chloride. The aerosol was drawn in through a sampling pump at 3 L min⁻¹, focused through an aerodynamic lens, and impacted on a collector plate kept at 600°C to vaporize the particles, and then ionized by electron ionization at 70 eV.



The ions were analyzed by a quadrupole mass spectrometer which provides unit mass resolution. The ACSM has an uncertainty of 30-40 % in mass concentration.

The ACSM was calibrated twice in the field using ammonium nitrate. A misaligned laser filament was replaced in January 2017, and the instrument was re-calibrated. A time- and composition-dependent collection efficiency (CDCE) corrects 1185 for the incomplete vaporization of mixed-phase particles (Middlebrook et al., 2012). The CDCE was calculated at each sampling interval using methods from Shilling and Levin (2021). The average CDCE between June and September was 0.54.

### 1.3 Single Particle Soot Photometer (SP2)

The single-particle soot photometer (SP2, Droplet Measurement Technology) measured time-dependent scattering and incandescence signals from refractory black carbon-containing particles with aerodynamic diameter between 80 and 500 nm. 1190 The SP2 used an 8-channel neodymium-doped yttrium–aluminum–garnet (Nd:YAG) laser (1064 nm) that induced incandescence of the particles to determine rBC mass and number concentrations (Stephens et al., 2003, Sedlacek 2017, ARM Handbook). The SP2 was calibrated twice in the field using fullerene soot with effective density from Gysel et al. (2011). The SP2 has a detection limit of 0.1 µg m$^{-3}$ with 75 % certainty (May et al., 2014) and a mass uncertainty of ~20 %.

### 1.4 Scanning Mobility Particle Sizer (SMPS)

The scanning mobility particle sizer (SMPS, TSI 3936) determines size distributions of number concentration of particles with mobility diameters between 10 and 1000 nm. The SMPS uses a bipolar aerosol charger to establish an equilibrium charge distribution on the aerosol. A long-column differential mobility analyzer (DMA) classifies charged particles according to their electrical mobility. The particle concentration was measured with a condensation particle counter (CPC). The SMPS and CPC were calibrated with certified polystyrene latex (PSLs), silver chloride, and sodium chloride 1200 particles (Hermann et al., 2007). The SMPS has an uncertainty of 10 %.

### 1.5 Particle Soot Absorption Photometer

The particle soot absorption photometer (PSAP, Radiance Research) measured optical transmittance through a cellulose filter on which particles were deposited to determine the light absorption at 470, 530, and 660 nm. PSAP data were corrected for loading and scattering effects following the ARM protocol ARM handbook (Virkkula 2010; Ogren 2010; Bond et al. 1999).

### 1.6 Nephelometer

The nephelometer (TSI, Inc) measured the total light scattering coefficient and backscatter fraction by aerosol particles at 450, 550, and 700 nm (Anderson and Ogren 1998). The aerosol optical properties and their derived products incorporate correction algorithms from Bond/Ogren (Bond 1999, extended by Ogren 2010) and Virkkula 2010, as done previously in Zuidema et al. (2018) and Pistone et al. (2019). Further information can be found in Flynn et al. (2020). The nephelometer



deployed at ASI was installed in series with a drying/humidification system to measure the change in scattering due to water

uptake or loss by the particles and has a systematic uncertainty of 10 % (Uin 2016b).

**1.7 CAPS-SSA**

The cavity attenuated phase shift single-scattering albedo monitor (CAPS-SSA, Aerodyne Inc) was deployed to

Ascension Island between August 4, 2017, and September 22, 2017. The CAPS-SSA collected 1 s time response measurements

of aerosol extinction and scattering. The CAPS-SSA measurement of ambient SSA at 530 nm wavelength compared well with

other optical measurements from the PSAP and nephelometers (Fig. A1) (Onasch et al., 2015).

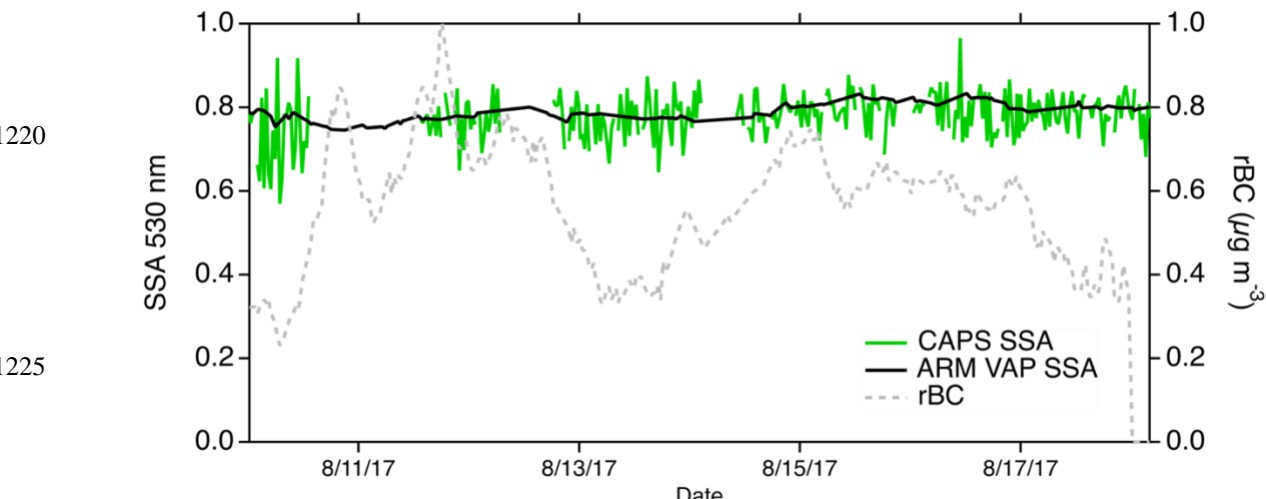

Figure A1. $SSA_{530}$ was determined from the PSAP and nephelometer (green) and CAPS-SSA (black). rBC mass concentration
(grey dashed line) indicates this comparison was done in the plume.

**1.8 CO/N$_2$O/H$_2$O Analyzer**

Carbon monoxide, nitrous oxide, and water vapor mixing ratios were measured with a CO/N$_2$O/H$_2$O Analyzer (Los

Gatos Research) at 1 s resolution. The measurement precision is 0.05 ppb under quiet ambient conditions (Springston, 2015).

ΔCO was calculated for each month by removing (subtracting) the 5[th] percentile of the measured CO from the total measured

CO. Background values of CO were 50.2 ppb for June, 54.4 ppb for July, 58.8 ppb for August, and 60.2 ppb for September.

Methods for calculating ΔCO were adopted from Che et al. (2022).

**2. Data analysis**

**2.1 Positive Matrix Factorization (PMF)**



A PMF apportions the unit mass resolution data using a bilinear model through a multilinear engine (ME-2). A graphical user interface, Source Finder (SoFi), developed at the Paul Scherrer Institute, was run using IGOR Pro (Wavemetrics v8) to facilitate the testing of different rotational techniques within the ME-2 (Canonaco et al., 2013; Crippa et al., 2014; Zhu et al., 2018; Canonaco et al., 2021). The PMF results in this paper were obtained by running 400-1000 clusters of the unit-mass resolution data from the 10 major plume events with SoFi through IGOR Pro (v8). The clusters were then analyzed using a k-

means statistical function to find clusters of comparable spatial extent. The average mass spectra result of the clusters presented two factors of source and process-related apportionments.

**2.2 SSA$_{530}$ and MAC$_{530}$ Calculations**

        The single-scattering albedo (SSA) is defined as the ratio of the (light) scattering coefficient to the sum of the absorption and scattering coefficients. The absorption Ångström exponent (AAE) describes the spectral dependence of light

absorption by aerosols (Helin et al., 2021) and is calculated from the linear fit of $\log(\sigma_a)$ to $\log(\lambda)$. The MAC$_{530}$ was calculated by dividing the absorption coefficient at 530 nm by the rBC mass concentration determined by the SP2.

        The Cavity Attenuated Phase Shift (CAPS) extinction monitor (Aerodyne Inc.) was deployed in August, 2017 to measure aerosol extinction (Sedlacek 2017). The calculated SSA$_{530}$ from the ARM VAP file agrees well with the SSA$_{530}$ from CAPS during smokey periods (Figure A1), indicating that the assumed corrections to the PSAP and nephelometers are

appropriate in the MBL where there could be a large presence of sea-salt particles.

**2.3 ACSM volume comparison**

The total volume concentration calculated using the size distribution from the SMPS was compared to the sum of the volume concentrations from the ACSM and SP2, calculated for each substance as the mass concentration divided by the density the bulk species based on the mass fraction of each species (OA, SO$_4$, NO$_3$, NH$_4$, rBC). The values of the densities of most of the

primary substances measured by the ACSM are well established (nitrate: $\rho = 1.72$ g cm$^{-3}$, sulfate and ammonium: $\rho = 1.77$ g cm$^{-3}$, and chloride: $\rho = 1.35$ g cm$^{-3}$; Nakao et al., 2013); however, values for OA between 0.8 and 1.8 g cm$^{-3}$ have been reported. Some of this range is due to differences in the chemical composition of the individual organic species and some from the age of the OA, with values of aged OA being typically less than those of freshly emitted OA (Rudich et al., 2006; Denjean et al., 2020b). Here a density of 0.9 g cm$^{-3}$ for OA was assumed, as aging metrics suggest that the aerosol at Ascension

Island is highly aged (greater than seven days old; Section 3.2.3). The volume concentration calculated as the sum of those determined by the SMPS agrees with that determined from the ACSM and SP2 to within 30 % (Fig. A2), which is roughly the sampling error expected for the latter two instruments.




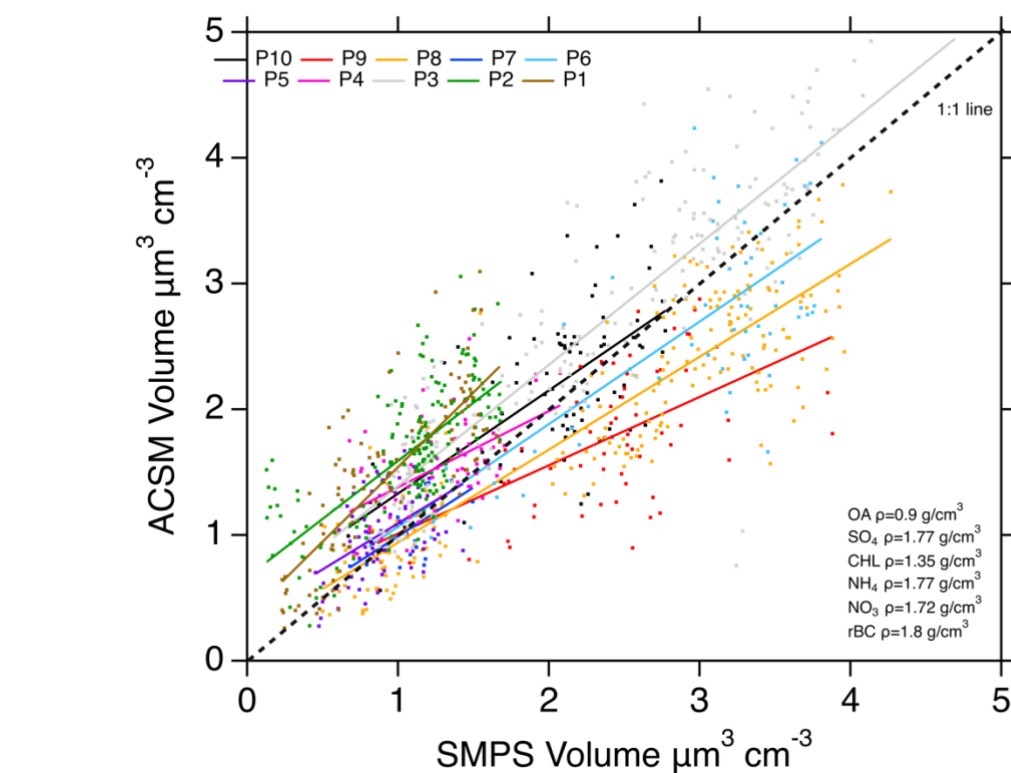

Figure A2. ACSM and SMPS volume comparison for the 10 selected plume events. Solid lines represent the best-fit linear regression analysis for each plume event. Lines and markers are colored by plume number. Black dashed line represents the 1:1 relationship.

**Appendix A References**

Anderson, T. L. and Ogren, J. A.: Determining Aerosol Radiative Properties Using the TSI 3563 Integrating Nephelometer, Aer. Sci. Tech., 29, 57–69, https://doi.org/10.1080/02786829808965551, 1998.

Bond, T. C., Anderson, T. L., and Campbell, D.: Calibration and Intercomparison of Filter-Based Measurements of Visible Light Absorption by Aerosols, Aerosol Science and Technology, 30, 582–600, https://doi.org/10.1080/027868299304435, 1999.

Canonaco, F., Crippa, M., Slowik, J. G., Baltensperger, U., and Prévôt, A. S. H.: SoFi, an IGOR-based interface for the efficient use of the generalized multilinear engine (ME-2) for the source apportionment: ME-2 application to aerosol mass spectrometer data, Atmos Meas Tech, 6, 3649–3661, https://doi.org/10.5194/amt-6-3649-2013, 2013.

Canonaco, F., Tobler, A., Chen, G., Sosedova, Y., Slowik, J. G., Bozzetti, C., Daellenbach, K. R., El Haddad, I., Crippa, M., Huang, R.-J., Furger, M., Baltensperger, U., and Prévôt, A. S. H.: A new method for long-term source apportionment with time-dependent factor profiles and uncertainty assessment using SoFi Pro: application to 1 year of organic aerosol data, Atmos Meas Tech, 14, 923–943, https://doi.org/10.5194/amt-14-923-2021, 2021.



Che, H., Segal-Rozenhaimer, M., Zhang, L., Dang, C., Zuidema, P., Sedlacek III, A. J., Zhang, X., and Flynn, C.: Seasonal variations in fire conditions are important drivers in the trend of aerosol optical properties over the south-eastern Atlantic, Atmos Chem Phys, 22, 8767–8785, https://doi.org/10.5194/acp-22-8767-2022, 2022.

Crippa, M., Canonaco, F., Lanz, V. A., Äijälä, M., Allan, J. D., Carbone, S., Capes, G., Ceburnis, D., Dall'Osto, M., Day, D.
A., DeCarlo, P. F., Ehn, M., Eriksson, A., Freney, E., Hildebrandt Ruiz, L., Hillamo, R., Jimenez, J. L., Junninen, H., Kiendler-Scharr, A., Kortelainen, A.-M., Kulmala, M., Laaksonen, A., Mensah, A. A., Mohr, C., Nemitz, E., O'Dowd, C., Ovadnevaite, J., Pandis, S. N., Petäjä, T., Poulain, L., Saarikoski, S., Sellegri, K., Swietlicki, E., Tiitta, P., Worsnop, D. R., Baltensperger, U., and Prévôt, A. S. H.: Organic aerosol components derived from 25 AMS data sets across Europe using a consistent ME-2 based source apportionment approach, Atmos Chem Phys, 14, 6159–6176, https://doi.org/10.5194/acp-14-6159-2014, 2014.

Flynn, C., Chand, D., Ermold, B., and Koontz, A.: The ARM Aerosol Optical Properties (AOP) The ARM Aerosol Optical Properties (AOP) Value-Added Product, November 2020.

Gysel, M., Laborde, M., Olfert, J. S., Subramanian, R., and Gröhn, A.: Effective density of Aquadag and fullerene soot black
carbon reference materials used for SP2 calibration, Atmos. Meas. Tech. Disc., 4, 4937–4955, https://doi.org/10.5194/amtd-4-4937-2011, 2011.

Hermann, M., Wehner, B., Bischof, O., Han, H.-S., Krinke, T., Liu, W., Zerrath, A., and Wiedensohler, A.: Particle counting efficiencies of new TSI condensation particle counters, J. Aerosol Sci., 38, 674–682,
https://doi.org/https://doi.org/10.1016/j.jaerosci.2007.05.001, 2007.

Kebabian, P. L., Robinson, W. A., and Freedman, A.: Optical extinction monitor using cw cavity enhanced detection, Rev. Sci. Instrum., 78, https://doi.org/https://doi.org/10.1063/1.2744223, 2007.

Onasch, T. B., Massoli, P., Kebabian, P. L., Hills, F. B., Bacon, F. W., and Freedman, A.: Single Scattering Albedo Monitor for Airborne Particulates, Aerosol Science and Technology, 49, 267–279, https://doi.org/10.1080/02786826.2015.1022248, 2015.

Ogren, J. A.: Comment on ``Calibration and Intercomparison of Filter-Based Measurements of Visible Light Absorption by
Aerosols'', Aerosol Science and Technology, 44, 589–591, https://doi.org/10.1080/02786826.2010.482111, 2010.

Leifer, R., R. H. Knuth, and H. N. Lee, 1994: Surface aerosol measurements at Lamont, Oklahoma. *Proc.* Third Atmospheric Radiation Measurement Science Team Meeting, Washington, DC, U.S. Department of Energy, 349–351.

Pistone, K., Redemann, J., Doherty, S., Zuidema, P., Burton, S., Cairns, B., Cochrane, S., Ferrare, R., Flynn, C., Freitag, S., Howell, S., Kacenelenbogen, M., LeBlanc, S., Liu, X., Schmidt, K. S., Sedlacek III, A. J., Segal-Rosenhaimer, M., Shinozuka, Y., Stamnes, S., van Diedenhoven, B., Van Harten, G., and Xu, F.: Intercomparison of biomass burning aerosol optical properties from in-situ and remote-sensing instruments in ORACLES-2016, Atmos. Chem. Phys., 19, 9181–9208, https://doi.org/10.5194/acp-19-9181-2019, 2019.

Shilling, J. E. and Levin, M. S.: Aerosol Chemical Speciation Monitor (ACSM) Composition-Dependent Collection Efficiency (CDCE) Value-Added Product Report, August 2021.

Sedlacek, A. J., 2016: Cavity attenuated phase shift (CAPS) monitor instrument handbook. ARM Tech. Rep. DOE/SC-ARM-
TR-155, 23 pp., https://doi.org/10.2172/1251390.

Sedlacek, A. J., 2017: Single-Particle Soot Photometer (SP2) instrument handbook. ARM Tech. Rep. DOE/SC-ARM-TR-169, 24 pp., https://doi.org/10.2172/1344179.



Springston, S. R., 2015: Carbon monoxide analyzer (CO-ANALYZER) instrument handbook. ARM Tech. Rep. DOE/SC-ARM-TR-159, 30 pp., https://doi.org/10.2172/1495422.

Stephens, M., Turner, N., and Sandberg, J.: Particle identification by laser-induced incandescence in a solid-state laser cavity, Appl. Opt., 42, 3726–3736, https://doi.org/https://doi.org/10.1364/AO.42.003726, 2003.

Uin, J., 2016: Integrating nephelometer instrument handbook. ARM Tech. Rep. DOE/SC-ARM-TR-165, 16 pp., https://doi.org/10.2172/1246075.

Uin, J., Aiken, A. C., Dubey, M. K., Kuang, C., Pekour, M., Salwen, C., Sedlacek, A. J., Senum, G., Smith, S.,

Wang, J., Watson, T. B., and Springston, S. R.: Atmospheric Radiation Measurement (ARM) Aerosol Observing Systems (AOS) for Surface-Based In Situ Atmospheric Aerosol and Trace Gas Measurements, J Atmos Ocean Technol, 36, 2429–2447, https://doi.org/10.1175/JTECH-D-19-0077.1, 2019.

Virkkula, A.: Correction of the Calibration of the 3-wavelength Particle Soot Absorption Photometer (3 PSAP), Aerosol Sci. Tech., 44, 706–712, 2010.

Watson, T. B., 2016b: Proton transfer time-of-flight mass spectrometer. *U* ARM Tech. Rep. DOE/SC-ARM-TR-160, 32 pp., https://doi.org/10.2172/1251396.

Zhu, Q., Huang, X.-F., Cao, L.-M., Wei, L.-T., Zhang, B., He, L.-Y., Elser, M., Canonaco, F., Slowik, J. G., Bozzetti, C., El-Haddad, I., and Prévôt, A. S. H.: Improved source apportionment of organic aerosols in complex urban air pollution using the multilinear engine (ME-2), Atmos Meas Tech, 11, 1049–1060, https://doi.org/10.5194/amt-11-1049-2018, 2018.

Zuidema, P., Sedlacek III, A. J., Flynn, C., Springston, S., Delgadillo, R., Zhang, J., Aiken, A. C., Koontz, A., and Muradyan, P.: The Ascension Island Boundary Layer in the Remote Southeast Atlantic is Often Smoky, Geophys. Res. Lett., 45, 4456–4465, https://doi.org/10.1002/2017gl076926, 2018.

**Data availability**

The following data are publicly available through the Atmospheric Radiation Measurement (ARM) Data Archive (for definitions see Table 1).

- ACSM: https://doi.org/10.5439/1762267 (Watson 2017; Shilling and Levin, 2021).
- SP2: http://www.arm.gov/campaigns/amf2016lasic/ (Sedlacek 2017)
- SMPS: https://doi.org/10.5439/1476898 (Kuang 2016)
- CPC: http://doi.org/10.5439/1352536 (Kuang 2016)
- Nephelometer: https://doi.org/10.5439/1369240 (Uin 2016)
- PSAP: https://doi.org/10.5439/1369240 (Springston 2018)
- Surface winds: https://doi.org/10.5439/1595321 (Holdridge 2020)
- CO analyzer: http://www.arm.gov/campaigns/amf2016lasic/ (Springston 2015)
- Optical properties VAP: https://doi.org/10.5439/1369240 (Flynn 2020)
- Fire data: https://firms.modaps.eosdis.nasa.gov/download/
- NCEP winds: https://psl.noaa.gov/data/gridded/data.ncep.reanalysis.html



- CAMS CO: https://ads.atmosphere.copernicus.eu/cdsapp#!/dataset/cams-global-reanalysis-eac4?tab=overview

**Supplement**


**Author contributions**

PZ designed the research. AD and PZ conceived this study. AJS and MZ prepared datasets of ACSM and SP2. AD, MZ, AJS analyzed datasets. AD, PZ, and EL wrote the paper with edits from TT, MZ, and AJS.

**Competing interests**

The contact author has declared that none of the authors has any competing interests.


**Acknowledgments**

We are grateful to the LASIC instrument mentors, AOS technicians, and logistics staff, who made this analysis possible through their efforts in deploying and maintaining the instruments at this remote location and to those who processed and calibrated the campaign data.


**Financial Support** A.D. and P.Z. were supported by DOE ASR award DE-SC0021250. E. R. L., A. J. S., and M. Z. were supported by the US Department of Energy's Office of Biological & Environmental Sciences (OBER) Atmospheric System Research (ASR) Program under contract DE-SC0012704 (the Process-level Advancements of Climate through Cloud and Aerosol Lifecycle Studies, PASCCALS)."

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
