# Peer review of "Burning conditions and transportation pathways determine biomass-burning aerosol properties in the Ascension Island marine boundary layer"

_EGUsphere, 2024_

## Referee Comment (RC1)

This study provides a detailed characterization of the chemical, physical and optical properties of transported biomass burning aerosols (BBAs) in the Ascension Island marine boundary layer (MBL), utilizing datasets from a long-term in-situ measurement platform. The authors present temporal variations of BBAs properties within the Ascension Island MBL, which are associated with variations in source burning conditions and transport pathways. The low single scattering albedo (SSA) values observed in this study are linked to low organic (OA) to black carbon (BC) ratios, attributed to OA loss as a consequence of oxidation processes. This finding underscores the importance of improving the representation of OA loss mechanisms and, consequently, aerosol optical properties in future modeling studies. I recommend publishing this work after addressing the corrections listed below.

General comments

1. Marine background environments over the Ascension Island have been considered when presenting chemical compositions such as $\Delta SO_4$. However, the marine background environments should be given more consideration when presenting the BBAs results such as FrBC and SSA.
   1) In this manuscript, the FrBC in the BBA was "calculated by dividing the number concentrations of rBC-containing particles measured by the SP2 by the total particle number concentration measured by the CPC" (lines 211-212). In the MBL over Ascension Island, the FrBC during BB-impacted periods is not only related to BBAs, but also likely affected by marine emissions. The concentration levels between background marine aerosols and mixed BBAs, as well as the mixing state (internally or externally) between BBAs and marine aerosols will affect the FrBC values.
   2) Similar to FrBC, the SSA given in this manuscript is also likely affected by such as marine sulfates in the MBL.
   The marine backgrounds may have small effects on these BBA properties, and the temporal trends shown in this manuscript may still depend on primarily difference in source conditions and transport processes. However, including the aforementioned details would provide a more comprehensive picture of BBAs in the MBL.

2. In Sect. 4.1, 4.4 and 5.1, the term of "SO$_4$:rBC" is used for discussions. However, based on descriptions in Sect 3, the term of SO$_4$:rBC should be changed to $\Delta SO_4$:rBC (refer to above-background $\Delta SO_4$ mass concentrations). For example, in lines 804, 809, 816 etc.

3. In Sect. 4.1, the authors state that (lines 824-825) "high OA:rBC, large rBC coating-to-core mass ratios, and high SO4:rBC would suggest that the fires were inefficient, *while the large values of rBC$_{gpd}$ and rBC$_{mpd}$ and high FrBC would indicate that the fires were efficient*". This assumes that efficient burning is related to larger BC core sizes. However, in the second paragraph of Sect. 4.2, the authors discuss that "the relationship between BC core size and source burning conditions are not sure from this study and previous studies".
   These present conflicting clarifications to some extent. I suggest rephrasing this part.

4. In Sect. 4 (line 755-757), the authors state that "Lastly, we examine the dependence of SSA$_{530}$ on the chemical properties of the BBA and explain why the BBA in the MBL has a lower SSA$_{530}$ than that in the FT", and further explanations follows in Sect. 4.5. I agree that the authors provide a detailed explanation on the relationship between SSA$_{530}$ and OA:rBC, combining the LASIC MBL and ORACLES FT. However, more details should be provided regarding the discussion of low SSA$_{530}$ in the Ascension Island MBL and the combination of different datasets:
   1) In line 998, the authors state that "The SSA$_{530}$ values in the Ascension Island MBL even lower than those in the nearby FT by about 0.07". However, it is not clearly demonstrated that what this "nearby FT" data is here. Does this "nearby FT data" refer to ORACLES FT? I suggest clarifying it clearer here.
   2) Following the above, the authors state that "This result is surprising, as mixing with clean marine air should increase SSA$_{530}$ in the MBL all else equal (Wu et al., 2020)".

In Barrett et al., (2022), LASIC ARM observed much less scatterings than CLARIFY FAAM during intercomparison flight legs, while the absorptions measurements were relatively consistent. This discrepancy in scattering measurements is attributed to the differences in the upper cut sizes of the impactors. LASIC scattering measurements are likely less affected by submicron coarse mode particles from marine sea sprays, due to its lower cut size ($0.78\,\mu m$) compared to CLARIFY ($1.0\,\mu m$). If the authors compare the LASIC-MBL/ORACLES-FT SSA trends to CLARIFY-MBL/FT SSA trends, the inconsistency in scattering measurements should be considered.

3) Is there any consistency or inconsistency in optical/chemical measurements between the LASIC and ORACLES platforms, if the authors combined two dataset to establish the correlation between $SSA_{530}$ and OA:rBC? E.g.

5. The uncertainties of calculated values from different instruments, such as SSA, should be provided in this manuscript.

6. When getting to Ascension Island, the clouds tend to be decoupled from the surface mixed layer (e.g. Abel et al., 2020), and the upper part of the MBL is relatively separated to the lower surface layer. Under this condition, surface layer measurements from LASIC may not directly represent BBAs properties below the cloud. This should be taken into account when interpreting the implications of LASIC datasets.

Specific comments

1. Page 1, Line 23: The authors state that "OA to rBC mass ratios (OA:rBC) in the MBL between 2 and 5 contrast to higher values of 5 to 15 observed in the nearby FT". I suggest adding the ages of BBAs for this MBL and "nearby FT".

2. Page 7, Line 200: What is the supporting material (reference or else) for using BC < 20 ng/m$^3$ as the criteria to calculate background $SO_4$ mass concentration?

3. Page 7, Table 2: Error with captions, is it $\Delta SO_4$ or $SO_4$? And the same question for CO, is it clean period CO or $\Delta CO$?

4. Page 8, Lines 225-226: What is the supporting material (reference or else) for using BC > 150 ng/m$^3$ as the threshold to define plume events?

5. Page 9: In Table 3, Regime 3 burning condition is described as "Inefficient and efficient".
Page 10, line 292-293: Regime 3 is described as "Although fires are still highly active in grassland regions during this time, these values indicate *inefficient combustion*".
Page 25: The reasons why Regime 3 had a mixture of inefficient and efficient combustion sources is explained in Sect. 4.1.
The description of Regime 3 burning condition is not consistent in Sect. 3 and 4, which may confuse the readers. I suggest rephrasing this part.

6. Pages 11-13, Figures 3-5: There are errors with figure caption, (b) should be surface relative humidity, and (c) should be land use map.

7. Page 14, Table 4: the last line of $SSA_{530}$, "0.81 0.01", a "$\pm$" is missing.

8. Page 16, Lines 499-502: The authors state that "the exception f44 values of P5 and P10… These differences possibly result from regime transitions and/or more complex fuel source mixtures". Could you provide more explanations why more complex fuel source mixtures may lead to the exception f44?

9. Page 18, Line 557: May need a reference for Hoppel minimum.

10. Page 18, Line 567-568: "The average of the bimodal size distributions during BBA-laden times had an Aitken mode with a peak 200 cm$^{-3}$ *at a larger diameter* near 45 nm". It should be clear to specify, "at a larger diameter" is larger than pristine period or?

11. Page 18, Line 584: "A lower fraction of the BBA-laden days (4 of 12) than *Regimes 1 and 2*".

12. Page 20, Line 628: "For example, the lowest SSA$_{530}$, 0.75±0.01, in P6, also had the highest FrBC". The highest FrBC is in P9, rather than the P6. Please rephrase this.

13. Page 20, Line 653: There is no Table S1 in the supplement. Please add it if it is needed.

14. Page 23, Line 744-745: There is a redundant reference of "(Ryoo et al., 2022)" in one sentence.

15. Page 24, Table 5: Wu et al. (2020) provided the SSA values at green channel. I suggest replacing SSA$_{658}$ with the blue channel SSA from Wu et al. (2020) in Table 5.

16. Page 26, Line 837: "(although still within instrument uncertainty)". What is the uncertainty of "rBC$_{mpd}$"?

17. Page 26, Lines 847-849: The authors state that "However, large rBC$_{gpd}$ and rBC$_{mpd}$ values of 135 and 205 nm, respectively, were also observed during Regime 3, in late August and early September when burning conditions were less efficient (rBC:ΔCO = 0.008)". The values (rBC:ΔCO = 0.008) for Regime 3 is not consistent with the average value (0.007) presented in Table 4.

18. Page 26, Lines 850-852: The authors state that "This contradicts this study's findings that rBC$_{gpd}$ and rBC$_{mpd}$ values were smaller (125 and 195 nm) when fires were also less efficient (rBC:ΔCO = 0.009)". Are these BC core size and rBC:ΔCO values from Regime 1 or which single plume event? I suggest clarifying this clearer.

19. Page 27-28, Lines 900-901: The authors state that "f44 values were greater than 0.22 and OA:rBC values were less than 5 for BBA sampled in the FT and MBL near Ascension Island". In Fig. 16, only the FT f44 values are presented. Is "MBL" near Ascension Island needed here?

20. Page 31, Line 1040: Error with the figure caption. a) SSA 530 nm versus "*FrBC*".

References:

Abel, S. J., Barrett, P. A., Zuidema, P., Zhang, J., Christensen, M., Peers, F., Taylor, J. W., Crawford, I., Bower, K. N., and Flynn, M.: Open cells exhibit weaker entrainment of free tropospheric biomass burning aerosol into the south-east Atlantic boundary layer, Atmos. Chem. Phys., 20, 4059–4084, https://doi.org/10.5194/acp-20-4059-2020, 2020.

Barrett, P. A., Abel, S. J., Coe, H., Crawford, I., Dobracki, A., Haywood, J. M., Howell, S., Jones, A., Langridge, J., McFarquhar, G., Nott, G., Price, H., Redemann, J., Shinozuka, Y., Szpek, K., Taylor, J., Wood, R., Wu, H., Zuidema, P., Bauguitte, S., Bennett, R., Bower, K., Chen, H., Cochrane, S. P., Cotterell, M., Davies, N., Delene, D., Flynn, C., Freedman, A., Freitag, S., Gupta, S., Noone, D., Onasch, T. B., Podolske, J., Poellot, M. R., Schmidt, S. K., Springston, S., III, A. J. S., Trembath, J., Vance, A., Zawadowicz, M., and Zhang, J.: Intercomparison of airborne and surface-based measurements during the CLARIFY, ORACLES and LASIC field experiments, Atmos. Meas. Tech., 15, 6329–6371, https://doi.org/10.5194/amt15-6329-2022, 2022.

Wu, H., Taylor, J. W., Szpek, K., Langridge, J. M., Williams, P. I., Flynn, M., Allan, J. D., Abel, S. J., Pitt, J., Cotterell, M. I., Fox, C., Davies, N. W., Haywood, J., and Coe, H.: Vertical variability of the properties of highly aged biomass burning aerosol transported over the southeast Atlantic during CLARIFY-2017, Atmos. Chem. Phys., 20, 12697–12719, https://doi.org/10.5194/acp-20-12697-2020, 2020.

---

## Author Comment (AC1)

We would like to sincerely thank the editor and reviewer for their time, effort, and thoughtful feedback on our manuscript. The reviewer comments are shown in black, with the author responses shown in blue and any edited manuscript language shown in *italicized blue font*.

General Comments

1. Marine background environments over the Ascension Island have been considered when presenting chemical compositions such as Δ$SO4$. However, the marine background environments should be given more consideration when presenting the BBAs results such as FrBC and SSA.

   - In this manuscript, the FrBC in the BBA was "calculated by dividing the number concentrations of rBC-containing particles measured by the SP2 by the total particle number concentration measured by the CPC" (lines 211-212). In the MBL over Ascension Island, the FrBC during BB-impacted periods is not only related to BBAs, but also likely affected by marine emissions. The concentration levels between background marine aerosols and mixed BBAs, as well as the mixing state (internally or externally) between BBAs and marine aerosols will affect the FrBC values.

   - Similar to FrBC, the SSA given in this manuscript is also likely affected by such as marine sulfates in the MBL.

The marine backgrounds may have small effects on these BBA properties, and the temporal trends shown in this manuscript may still depend on primarily difference in source conditions and transport processes. However, including the aforementioned details would provide a more comprehensive picture of BBAs in the MBL.

The reviewer brings up a good point here. The aerosol size distributions in Figure 11 nicely demonstrate how the accumulation mode number concentration differs between clean and polluted time periods. We have added additional language to inform the reader that background aerosol concentrations in the accumulation mode are a fraction of those during the biomass burning plume events. We are only focusing on accumulation mode aerosols here because the SP2 cannot sample particles < 80 nm. SSA$_{530}$ values are also not likely affected by the small contribution of background aerosol particles in either the accumulation or Aitken mode during the plume events. We have added additional language to Section 4.5. The mean SSA$_{530}$ values during clean events was 0.98±0.01 and highlight how the BBA substantially lowers the SSA.

We have added the following text to Section 2.2, *"The FrBC values are not likely influenced by the background aerosol emissions because aerosol number concentrations in the accumulation mode during these clean periods is only a fraction of the BBA aerosol number concentrations sampled during plume events. We further highlight these differences in aerosol number concentration between the clean and polluted periods in Sect. 3.2.3."*

We have added the following text to Section 4.5, *"This result is surprising, as the mean background SSA$_{530}$ value was 0.98±0.01, and mixing with clean marine air should increase SSA$_{530}$ in the MBL. Previous studies, such as Wu et al. (2020), concluded that BBA in the MBL is*

*less absorbing due to this mixing. In contrast, our results demonstrate that BBA in the MBL is more absorbing than that in the FT."*

2. In Sect 4.1, 4.4 and 5.1, the term of "SO4:rBC" is used for discussions. However, based on descriptions in Sect 3, the term of SO4:rBC should be changed to Δ*SO4:rBC* (refer to above-background ΔSO4 mass concentrations). For example, in lines 804, 809, 816 etc.

The text in Sections 4.1, 4.4. and 5.1 where SO₄ and SO₄:rBC are referenced has been corrected to ΔSO₄ and ΔSO4:rBC throughout. Thank you for catching this typo.

3. In Sect. 4.1, the authors state that (lines 824-825) "high OA:rBC, large rBC coating-to-core mass ratios, and high SO4:rBC would suggest that the fires were inefficient, *while the large values of rBCgpd and rBCmpd and high FrBC would indicate that the fires were efficient*". This assumes that efficient burning is related to larger BC core sizes. However, in the second paragraph of Sect. 4.2, the authors discuss that "the relationship between BC core size and source burning conditions are not sure from this study and previous studies".

These present conflicting clarifications to some extent. I suggest rephrasing this part.

We have adjusted the text in Section 4.1 to exclude any discussion of the rBC size distributions. The reviewer makes a good point that the previous language was confusing and conflicted with Section 4.2. We kept Section 4.2 as is because it provides a more thorough discussion on the rBC size distributions.

4. In Sect. 4 (line 755-757), the authors state that "Lastly, we examine the dependence of SSA530 on the chemical properties of the BBA and explain why the BBA in the MBL has a lower SSA530 than that in the FT", and further explanations follows in Sect. 4.5. I agree that the authors provide a detailed explanation on the relationship between SSA530 and OA:rBC, combining the LASIC MBL and ORACLES FT. However, more details should be provided regarding the discussion of low SSA530 in the Ascension Island MBL and the combination of different datasets:
   - In line 998, the authors state that "The SSA530 values in the Ascension Island MBL even lower than those in the nearby FT by about 0.07". However, it is not clearly demonstrated that what this "nearby FT" data is here. Does this "nearby FT data" refer to ORACLES FT? I suggest clarifying it clearer here.

We have adjusted the following text in Section 4.5 to clarify what region we are referring to.
*"These values are even lower (by about 0.07) than those sampled in the FT between the African plateau and Ascension Island during the September ORACLES 2016 campaign."*

   - Following the above, the authors state that "This result is surprising, as mixing with clean marine air should increase SSA530 in the MBL all else equal (Wu et al., 2020)".

Wu et al., 2020 conclude that the higher SSA values in the MBL are due to the BBA mixing with the clean marine air, especially near the surface layer. The results presented here, show that

during biomass burning events, the BBA dominates the aerosol optical properties and not submicron sea salt.

From Comment 1 above: We have added the following text to Section 4.5, *"This result is surprising, as the mean background $SSA_{530}$ value was $0.98\pm0.01$, and mixing with clean marine air should increase $SSA_{530}$ in the MBL. Previous studies, such as Wu et al. (2020), concluded that BBA in the MBL is less absorbing due to this mixing. In contrast, our results demonstrate that BBA in the MBL is more absorbing than that in the FT."*

In Barrett et al., (2022), LASIC ARM observed much less scatterings than CLARIFY FAAM during intercomparison flight legs, while the absorptions measurements were relatively consistent. This discrepancy in scattering measurements is attributed to the differences in the upper cut sizes of the impactors. LASIC scattering measurements are likely less affected by submicron coarse mode particles from marine sea sprays, due to its lower cut size ($0.78 \mu m$) compared to CLARIFY ($1.0 \mu m$). If the authors compare the LASIC-MBL/ORACLES-FT SSA trends to CLARIFY-MBL/FT SSA trends, the inconsistency in scattering measurements should be considered.

- Is there any consistency or inconsistency in optical/chemical measurements between the LASIC and ORACLES platforms, if the authors combined two dataset to establish the correlation between SSA530 and OA:rBC? E.g.

The aerosol optical instruments deployed during LASIC were behind a 1µm impactor (Uin et al., 2020) which is identical to the CLARIFY 1 µm cut off. The $SSA_{530}$ measurements calculated in the ARM VAP (from the PSAP and the nephelometer) compare well to those from the $CAPS_{SSA}$ monitor that was also deployed to Ascension Island (Fig. A2). The consistency between the ARM VAP $SSA_{530}$ values and the $CAPS_{SSA}$ values when BBA is present provides confidence that the scattering and absorption coefficients measured at Ascension Island during LASIC are trustworthy. However, we cannot explain the discrepancies in the scattering coefficient between the LASIC and CLARIFY described in Barrett et al. (2022).

The submicron $SSA_{530}$ calculations from ORACLES were performed with the same set of corrections (Pistone et al., 2019; Dobracki et al., 2023) that were used in the ARM VAP (Flynn et al., 2020; Kassianov et al., 2023). The ORACLES campaign also deployed the same models of the PSAP (Radiance Research) and Nephelometer (TSI 3563). Therefore, we expect consistency between the ORACLES and LASIC optical property measurements.

We have added the following text to Section 2.1, *"The aerosol optical properties measured in this study were also compared to those from the ORACLES and CLARIFY campaigns. During ORACLES, the aerosol optical properties were measured using the same instruments as in the LASIC campaign. Specifically, scattering coefficients at wavelengths of 450, 550, and 700 nm were measured using the TSI 3563 nephelometer, while absorption coefficients at 465, 530, and 650 nm were measured with the PSAP (Radiance Research) (Dobracki et al., 2023). Corrections for the scattering coefficients were applied using both Virkkula and Bond & Ogren methods (Pistone et al., 2019; Dobracki et al., 2023). In contrast, during the CLARIFY campaign, absorption coefficients at 405, 515, and 660 nm, along with aerosol dry extinction at 405 and 658 nm, were measured using the EXtinction SCattering and Absorption of Light for Airborne Aerosol Research (EXSCALABAR) (Wu et al., 2020; Barrett et al., 2022). Despite these*

*differences in measurement techniques, we aim to utilize the data from both campaigns for comparative analysis."*

Regarding the chemical properties, there were slight differences between the mass spectrometer and the SP2 models. The ACSM at Ascension Island is the lower resolution model of the Aerosol Mass Spectrometer (AMS) that was used during ORACLES. However, because we are using the bulk chemical composition and comparing the OA:rBC ratios (not absolute concentrations of OA $\mu g\ m^{-3}$), there should be no major inconsistencies due to collection methods or instrument abilities. The ACSM and AMS data were both corrected with the composition dependence collection efficiency (Dobracki et al., 2023).

The SP2 at Ascension Island was the 8-channel model from DMT, whereas the SP2 on ORACLES was the 4-channel model from DMT. The advantage of the 8-channel model is that it can provide the coating thickness. Aside from that feature, the rBC mass measurements from both LASIC and ORACLES should be consistent as the data was collected and processed by Arthur J. Sedlacek (BNL).

We have added the following text to Section 2.1, *"Non-refractory aerosols during the ORACLES 2016 campaign were sampled using a High-Resolution Time-of-Flight Aerosol Mass Spectrometer (HR-ToF-AMS, Aerodyne Inc.)(Dobracki et al., 2023). During the CLARIFY campaign, the same type of aerosol was sampled with a Compact Time-of-Flight Aerosol Mass Spectrometer (C-ToF-AMS) (Taylor et al., 2020; Barrett et al., 2022). rBC sampled during the ORACLES 2016 campaign and CLARIFY campaign was measured with a four-channel Single Particle Soot Photometer (SP2, DMT) (Barrett et al., 2022; Sedlacek et al., 2022). Despite these slight differences in instrumentation and sampling strategies, we utilize the data from both campaigns in this study to ensure a comprehensive analysis."*

5.  The uncertainties of calculated values from different instruments, such as SSA, should be provided in this manuscript.
We have added a column to Table 2 listing the instrument or calculation uncertainty when applicable.

6.  When getting to Ascension Island, the clouds tend to be decoupled from the surface mixed layer (e.g. Abel et al., 2020), and the upper part of the MBL is relatively separated to the lower surface layer. Under this condition, surface layer measurements from LASIC may not directly represent BBAs properties below the cloud. This should be taken into account when interpreting the implications of LASIC datasets.

This is a fair point, however the aircraft measurements bias the conditions. Zhang and Zuidema (2019) report that the clouds at Ascension Island are coupled with surface during the early morning and evening, especially in August. The decoupling typically occurred during the day, when the aircrafts sampled near Ascension Island. It would be more appropriate to say that the clouds and the surface were intermittently coupled. We do agree that additional language should be added to Section 2.1. We have added the following text, *"While the LASIC data are constrained to surface based measurements, they are predominantly representative of the*

*boundary layer surrounding Ascension Island. It is crucial to consider that the clouds intermittently coupled with the surface during the early morning and evening. The observed intermittent coupling of clouds with the surface during early morning and evening hours highlights a more complex interaction than the constant decoupling suggested by aircraft measurements (Zhang and Zuidema., 2019). Despite these complexities, our findings provide a reliable representation of the prevailing boundary layer conditions (Zhang and Zuidema, 2019; Abel et al., 2020; Wu et al., 2020)."*

Specific Comments

1. Page 1, Line 23: The authors state that "OA to rBC mass ratios (OA:rBC) in the MBL between 2 and 5 contrast to higher values of 5 to 15 observed in the nearby FT". I suggest adding the ages of BBAs for this MBL and "nearby FT".

We have adjusted the text to read *"OA to rBC mass ratios (OA:rBC) in the MBL between 2 and 5 for BBA aged over 10 days contrast to higher values of 5 to 15 for BBA aged between 4 and 8 days observed in the nearby FT."*

2. Page 7, Line 200: What is the supporting material (reference or else) for using $BC < 20$ ng/m3 as the criteria to calculate background SO4 mass concentration?

This was the 5th percentile of the rBC mass concentrations for June, July, August, and September. This method was adopted from Che et al., (2022). This detail has been added to the text the sentence now reads *"…when rBC mass concentrations were less than 20 ng m$^{-3}$, the bottom 5th percentile…"*

3. Page 7, Table 2: Error with captions, is it $\Delta SO_4$ or $SO_4$? And the same question for CO, is it clean period CO or $\Delta CO$?

The table caption and the text contained within the table have been corrected to $\Delta SO_4$ and $\Delta CO$.

4. Page 8, Lines 225-226: What is the supporting material (reference or else) for using $BC > 150$ ng/m3 as the threshold to define plume events?

This was the 70th percentile rBC mass concentration for June, July, August, and September. This value is double the threshold that Che et al., (2020) used to define plume events. This threshold was to ensure that BBA dominated the plume events.

5. Page 9: In Table 3, Regime 3 burning condition is described as "Inefficient and efficient". Page 10, line 292-293: Regime 3 is described as "Although fires are still highly active in grassland regions during this time, these values indicate *inefficient combustion*". Page 25: The reasons why Regime 3 had a mixture of inefficient and efficient combustion sources is explained in Sect 4.1. The description of Regime 3 burning condition is not consistent in Sect. 3 and 4, which may confuse the readers. I suggest rephrasing this part.

We have corrected the text in Section 3.1 to avoid confusion between the two sections. The paragraph in Section 3.1 now reads, *"The final two plume events, spanning August 24 to September 11, have the lowest overall rBC:ΔCO, with a mean of 0.0071±0.0004. These low rBC:ΔCO values indicate that the fires were inefficient, however, most fires occurred east of*

*30˚E and south of 10˚S (Fig. 5a, in northeast Zambia, southwest Tanzania, Mozambique, and Zimbabwe), with surface RH ranging between 30 and 60 % (Fig. 5b), over vegetation types varying from grasslands to woody savannas (Fig. 5c). Also, most of the fires occurred over dry central Africa and many also occurred on the eastern African coast where precipitation was greater in September (Ryoo et al., 2021). The variation in surface RH, and vegetation suggest that the two plumes in this regime originated from fires that are both efficient and inefficient. We further describe these conditions in Section 4. A notable feature of this regime is that the strong free-tropospheric winds known as the African Easterly Jet-South became active around August 20, at approximately 700 hPa (Ryoo et al., 2022)."*

6. Pages 11-13, Figures 3-5: There are errors with figure caption, (b) should be surface relative humidity, and (c) should be land use map.
This has been corrected.

7. Page 14, Table 4: the last line of SSA530, "0.81 0.01", a "±" is missing.
This has been corrected.

8. Page 16, Lines 499-502: The authors state that "the exception f44 values of P5 and P10… These differences possibly result from regime transitions and/or more complex fuel source mixtures". Could you provide more explanations why more complex fuel source mixtures may lead to the exception f44?

The BBA over the SEA is some of the most oxidized aerosol that has been intensively sampled. During the PMF analysis, we could barely distinguish two factors, as the organic aerosol was mostly homogeneous by the time it reached Ascension Island because it was so aged. Typically, PMF analyses apportion up to 5 aerosol sources because most field campaigns study more heterogeneous organic aerosols. We lack sufficient information to determine the causes of these slight differences in mass spectrums due to the limited amount of PMF data on highly oxidized organic aerosol. Ultimately, we took the average of the *f44* values to make help ease the interpretation of the PMF analysis.

9. Page 18, Line 557: May need a reference for Hoppel minimum.
We have added the reference (Hoppel et al., 1986) here.

10. Page 18, Line 567-568: "The average of the bimodal size distributions during BBA-laden times had an Aitken mode with a peak 200 cm-3 *at a larger diameter* near 45 nm". It should be clear to specify, "at a larger diameter" is larger than pristine period or?
We have rewritten the sentence as, *"The average of the bimodal size distributions during BBA-laden times had an Aitken mode with a peak 200 cm$^{-3}$ at a larger diameter than that during the pristine period, near 45 nm, and an accumulation mode with a much larger peak than that during the pristine period ranging from 400-700 cm$^{-3}$ at a diameter near 165 nm (Fig. 11a), with a Hoppel minimum diameter near 70 nm."*

11. Page 18, Line 584: "A lower fraction of the BBA-laden days (4 of 12) than *Regimes 1 and 2*".

We have rewritten the sentence as, *"A lower fraction of the BBA-laden days (4 out of 12) than in Regimes 1 and 2 had bimodal number size distributions (Fig. S4c)…"*

12. Page 20, Line 628: "For example, the lowest SSA530, 0.75□0.01, in P6, also had the highest FrBC". The highest FrBC is in P9, rather than the P6. Please rephrase this.

We have rewritten the sentence as, *"Lower $SSA_{530}$ values were typically associated with higher FrBC and lower OA:rBC and $\Delta SO_4$:rBC. For example, the lowest $SSA_{530}$, $0.75\pm0.01$, in P6, also had the second highest FrBC ($0.32\pm0.04$)..."*

13. Page 20, Line 653: There is no Table S1 in the supplement. Please add it if it is needed.

The sentence was from a previous draft and has been removed. The current draft provides greater detail about the transportation pathways than the previous draft. Therefore, a supplemental table is no longer needed.

14. Page 23, Line 744-745: There is a redundant reference of "(Ryoo et al., 2022)" in one sentence.

We have corrected the sentence. *"The African Easterly Jet South became active after Aug. 20 causing a dramatic switch in the BBA transport to much higher in the FT, near 700 hPa (Ryoo et al., 2022)."*

15. Page 24, Table 5: Wu et al. (2020) provided the SSA values at green channel. I suggest replacing SSA658 with the blue channel SSA from Wu et al. (2020) in Table 5.

We chose to use the SSA values at 658 nm because those are shown in Figure 7 for the polluted boundary layer (Wu et al., 2020) and that time period (Period 3) is the most relevant to the results we are presenting in this study. We acknowledge that Wu et al., (2020) states that the SSA values at 550 nm ranged from 0.88 to 0.97; however, this range appears to be the range for the entire study, not just the polluted periods. We would prefer to keep the values in the table as is. We have added text to Section 2.5 briefly describing differences in optical property measurements (see Comment 4 above).

16. Page 26, Line 837: "(although still within instrument uncertainty)". What is the uncertainty of "rBCmpd"?

The uncertainty of the SP2 was ~20% (line 256).

17. Page 26, Lines 847-849: The authors state that "However, large rBCgpd and rBCmpd values of 135 and 205 nm, respectively, were also observed during Regime 3, in late August and early September when burning conditions were less efficient (rBC:ΔCO = 0.008)". The values (rBC:ΔCO = 0.008) for Regime 3 is not consistent with the average value (0.007) presented in Table 4.

The text has been corrected to agree with the table. The rBC:ΔCO value of 0.007 is correct.

18. Page 26, Lines 850-852: The authors state that "This contradicts this study's findings that rBCgpd and rBCmpd values were smaller (125 and 195 nm) when fires were also less efficient (rBC:$\Delta$CO = 0.009)". Are these BC core size and rBC:$\Delta$CO values from Regime 1 or which single plume event? I suggest clarifying this clearer.

We have rewritten the sentence as *"Yet, smaller $rBC_{gpd}$ and $rBC_{mpd}$ values in Regime 1 (125 and 195 nm) occurred when fires were also less efficient (rBC:$\Delta$CO=0.008), which is contradictory to the higher values of $rBC_{gpd}$ and $rBC_{mpd}$ in Regime 3 that support Holder et al., (2016). These contrasting results suggest that further research is needed on the dependence of rBC core diameters on burning conditions and fuel types."*

19. Page 27-28, Lines 900-901: The authors state that "f44 values were greater than 0.22 and OA:rBC values were less than 5 for BBA sampled in the FT and MBL near Ascension Island". In Fig. 16, only the FT f44 values are presented. Is "MBL" near Ascension Island needed here?

The green marker on Figure 16 shows the LASIC MBL f44 and OA:rBC values. We have added an orange marker for CLARIFY data in the MBL based on the results from Wu et al., (2020).

The figure caption now reads *"Figure 16. OA:rBC versus f44 for the FT during ORACLES (2016-2018)(red), CLARIFY FT (2017) (blue) (Wu et al., 2020), CLARIFY 2017 MBL (Wu et al., 2020)(orange), ATTO (2014)(black) (Holanda et al., 2020), and for the LASIC MBL (2017) (green dash). Error bars represent the standard deviation of the data set."*

20. Page 31, Line 1040: Error with the figure caption. a) SSA 530 nm versus "*FrBC*".

The figure caption now reads *" a) $SSA_{530}$ versus FrBC"*

---

## Author Comment (AC2)

We thank the reviewer for their thoughtful comments and insights on our manuscript. We appreciate the time and effort that was dedicated to providing these suggestions. These revisions have enhanced the clarity and impact of our work. The reviewer comments are shown in black, with the author responses shown in blue and any edited manuscript language shown in *italicized blue font*.

The main major comment I have on this paper is regarding the Regime separation and its justification and significance. The authors state that the division is supported by differing burn conditions (fire density, surface RH, land use, and fuel type) over the season (Line ~228+, Table 3), but it is difficult to see how these differences are supported in the analysis (Figs 2; 3-5).

Due to limited availability of conserved tracer data (i.e., $CO_2$ and other gaseous tracers) and airspace restrictions, we included reanalysis data to aid in interpreting the burning conditions over land. This approach aligns with Che et al. (2022), who similarly used soil moisture content as a supplementary metric to characterize burning conditions throughout the season. We have revised the text in Section 2.2 to clarify that the reanalysis data serve as supplementary information for interpreting the rBC:ΔCO ratios rather than as a primary classification criterion.

*"Surface RH fields provided by the NOAA National Center for Environmental Prediction (NCEP) reanalysis are robustly used to assess the burning condition classification, with RH values >50% indicating efficient fires and <50% indicating inefficient fires. The surface RH analysis serves as supplementary information for interpreting the rBC:ΔCO ratios, rather than as a primary classification criterion, similar to the approach used in Che et al. (2022a). Inefficient fires typically produce relatively more OA and $SO_4$ and relatively less rBC than efficient fires (Collier et al., 2016; Rickly et al., 2022). Although modified combustion efficiency (MCE) may be a better determinant of burning conditions (Collier et al., 2016; Dobracki et al., 2023), this quantity could not be calculated because CO2 was not sampled at Ascension Island during the LASIC campaign."*

The rBC:delCO distributions are described in the text (e.g. Line 16) as being used to determine the different fuel and burn conditions, but as these values are shown in Fig 2, they are not obviously subdivided according to a particular criterion

We note here that the classification of the 10 plumes into the 3 regimes is mostly meant to aid interpretation. The regime differentiation is not statistically significant. Indeed, one remarkable feature of the long-range transport BBA is how little it changes, with PMF analyses only identifying two major components that mostly resemble LVOOA. Previous studies from LASIC have often classified BBA using monthly-means (Zuidema et al., 2018; Carter et al., 2021; Che et al., 2022) However, smoke loading at Ascension is strongly synoptically modulated (evident in Fig. 1 of this manuscript) with very clean time periods interspersed with time periods experiencing significant loading. Thus, monthly-means (which are inherently another arbitrary time unit separately by only one

day) may not adequately capture the BBA properties in the Ascension Island marine boundary layer (MBL).

We have added the following text to Section 3. "*Previous studies from LASIC have often classified BBA using monthly means (Zuidema et al., 2018; Carter et al., 2021; Che et al., 2022). However, due to the variability in aerosol loading across the four months, monthly means may not adequately capture the BBA properties in the Ascension Island MBL. Here we explore ten plume events…*"

This is evident simply by reconsidering the grouping of rBC:Delta_CO ratios by month, (shown below) with the red dashed lines now distinguishing the June, July, August and September smoke plumes.

[Figure]

As shown, the rBC:Delta_CO ratios indicate the first plume event in July (P5) more closely resembles the June BBA characteristics, the first plume event in August is more similar to late July BBA, and the last plume in August was more similar to the September BBA. Given that we wanted to move away from monthly-means to a more individual analysis, when differences in the BBA characteristics are nevertheless subtle compared to more complex northern hemisphere urban environments, yet also wanted to reduce the complexity of 10 individual events to a fewer number, we came up with the regime discrimination we used throughout the manuscript.

In response to the reviewer's feedback, we did revise the manuscript to reduce the emphasis on reanalysis data (RH, fire density, land use) while retaining the regime classification based on the rBC:ΔCO ratio.

In Section 2.2, we have included the following text, "*Surface RH fields provided by the NOAA National Center for Environmental Prediction (NCEP) reanalysis are robustly used to assess the burning condition classification, with RH values >50% indicating efficient fires and <50% indicating inefficient fires. The surface RH analysis serves as supplementary information for interpreting the rBC:ΔCO ratios, rather than as a primary classification criterion, similar to the approach used in Che et al. (2022a). Inefficient fires typically produce relatively more OA and SO₄ and relatively less rBC than efficient fires (Collier et al., 2016; Rickly et al., 2022). Although modified combustion efficiency (MCE) may be a better determinant of burning conditions (Collier et al., 2016; Dobracki et al., 2023), this quantity could not be calculated because CO2 was not sampled at Ascension Island during the LASIC campaign.*"

And in Section 3, we have included the following text, "*Previous studies from LASIC have often classified BBA using monthly means (Zuidema et al., 2018; Carter et al., 2021; Che et al., 2022). However, due to the variability in aerosol loading across the four months, monthly means may not adequately capture the BBA properties in the Ascension Island MBL. Here we explore ten plume events…*"

Please note, these are the same responses from the first two comments above.

Particularly looking at plume P2, it doesn't seem to "suggest burning conditions remained mostly homogeneous over the six weeks" (Line 281). I wonder if there's any evidence to suggest that P2 is more in line with P9-10 in terms of origins? Figs 9, 10, and 12 also don't clearly show 3 distinct sets of properties, so I would like to see a more concrete justification for that division, if you choose to keep it. (and actually, P2 is an obvious outlier in Fig 12 as well).

As also in the above figure, P2 is clearly an outlier compared to its neighboring-in-time smoke plumes. We have provided additional text in Section 3.1 to highlight this anomalous plume.

Lines 353-354 *"Despite the anomalously low rBC:ΔCO value of 0.006±0.001 in P2, the rBC:ΔCO ratios of the other four plumes suggest that burning conditions remained mostly homogeneous over the six weeks."*

Lines 362-380 *"The final two plume events, spanning August 24 to September 11, have the lowest overall rBC:ΔCO, with a mean of 0.0071±0.0004. These low rBC:ΔCO values indicate that the fires were inefficient, however, most fires occurred east of 30˚E and south of 10˚S (Fig. 5a, in northeast Zambia, southwest Tanzania, Mozambique, and Zimbabwe), with surface RH ranging between 30 and 60 % (Fig. 5b), over vegetation types varying from grasslands to woody savannas (Fig. 5c). Also, most of the fires occurred over dry central Africa and many also occurred on the eastern African coast where precipitation was greater in September (Ryoo et al., 2021). The variation in surface RH, and vegetation insinuate that the two plumes in this regime may have originated from fires that are both efficient and inefficient. We further describe these*

*conflicting conditions in Section 4. A notable feature of this regime is that the strong free-tropospheric winds known as the African Easterly Jet-South became active around August 20, at approximately 700 hPa (Ryoo et al., 2022).*

Additional daily HYSPLIT back trajectories reveal that this plume likely originated from northwestern Angola and little to no precipitation occurred as the plume was transported to Ascension Island. This is now discussed in Section 3.3.

*"Additional HYSPLIT back trajectories (not shown) highlight that P2 with the anomalously low rBC:ΔCO ratio (0.006±0.001) also originated from northern Angola, which suggests that the source of this plume was likely similar to those from the other four plume events in this regime."*

[Figure]

The source origin of P2 is similar to P1, P3-P5, and there is no evidence that precipitation removed rBC (while maintaining the CO). While the rBC:ΔCO ratio aligns more closely with that from Regime 3, the source region of P2 aligns more closely with that from Regime 1. The rBC core size and the aerosol number size distributions from P2 also more closely align with the other four plumes in Regime 1 (Figs. 6, 11a). Overall, we can only conclude that perhaps this BBA originated from a late-stage fire, when the most intense flaming was over, and was developing characteristics of a more smoldering fire. This would explain why this plume had lower rBC mass concentrations than the other plumes in Regime 1 relative to OA, $SO_4$, and CO, likely due to less-efficient combustion processes. This conjecture must remain speculative but is consistent with the measurements we do have. The reanalysis and HYSPLIT back-trajectories indicate P2 should be viewed in the context of the neighboring-in-time plumes, and not grouped with P9-P10.

We also respond to the further concern raised by the reviewer, that the transition between Regime 1 and 2, Fig 3 vs Fig 4 show averages over two periods of 4-6 weeks each, with only one day of separation between them. It's not at all obvious that e.g. the spatial distribution of fire density is meaningfully different between these two periods (Table 4 also seems to indicate that many parameters may be statistically indistinguishable between the two of them); instead, these figures' panels (a) seem to show that there is greater density in the latter period, but that the fires occur over largely the same spatial domain. A distinction of surface RH <50% or >50% (p. 10) also seems a bit arbitrary a cutoff, and e.g. it's not actually clear that the locations of the fires in Fig 3 correspond to RH>50% (line 283).

As mentioned above, we have revised the text in Section 2.2 and placed less emphasis on the reanalysis data.

"*Surface RH fields provided by the NOAA National Center for Environmental Prediction (NCEP) reanalysis are robustly used to assess the burning condition classification, with RH values >50% indicating efficient fires and <50% indicating inefficient fires. The surface RH analysis serves as supplementary information for interpreting the rBC:ΔCO ratios, rather than as a primary classification criterion, similar to the approach used in Che et al. (2022a). Inefficient fires typically produce relatively more OA and $SO_4$ and relatively less rBC than efficient fires (Collier et al., 2016; Rickly et al., 2022). Although modified combustion efficiency (MCE) may be a better determinant of burning conditions (Collier et al., 2016; Dobracki et al., 2023), this quantity could not be calculated because $CO_2$ was not sampled at Ascension Island during the LASIC campaign.*"

Please note this is the same response from the first comment above.

The reanalysis data was used as a tool to help with interpretation of the large LASIC data set. We wanted to move away from a monthly mean analysis, which also would have had 1 day of separation between observational periods. Previous studies have shown that as the biomass burning season progresses, the grass (moisture) content decreases (Hoffa et al., 1999; Korontzi et al., 2003 ) and the fires shift toward central and southeast Africa

(Redemann et al., 2022). The surface RH and fire density maps in Figures 3-5 corroborate this.

In the same vein, the time intervals averaged in Figures 3-5, if I'm reading it correctly, show the continental conditions directly coincident with the observed conditions at ASI. Yet, obviously, airmasses don't arrive at ASI instantaneously; according to Figs 13 and 19, the transport time to reach ASI is ~5 days minimum and may be even greater than 10 days, so do the continental conditions during the exact same time actually indicate changes in conditions? And if so, shouldn't the conditions (and Regime definition) be lead/lagged relative to the ASI observations?

We have updated figures 3-5 to include fire density and surface RH data 7 days prior to the start of each regime. We had already accounted for this in Figures 12 and 13. And no, we are basing the regime definition on the aerosol characteristics perceived at Ascension by design because the rBC:ΔCO ratio is conserved on these timescales.

This is also complicated by Table 2 vs Table 3; the Regimes are defined as these larger periods, and yet there are also "clean periods" within those periods. So do Figs 3-5 show the average including "clean" times? From Fig 1 it's about half clean, half plume over a given "regime," so even if transport time lag is taken into account, did the fires vary within these longer periods?

The clean periods at Ascension Island are due to the presence of a sea-level pressure high between 0° and 20°W, which promoted the advection of pristine air from the southern oceans and forced the BBA to north of Ascension Island. Though the smoke plumes did not reach Ascension Island, there were still fires over these time periods which is why they are included in Figures 3-5. The values in Table 3 only include data from the plume. We have added a footnote on Table 3 to clarify this.

The "efficient"/"inefficient" burning condition distinction also was not clear: the low rBC:delCO mass ratios = inefficient combustion (Line 280, 806+); higher values in the second period = efficient combustion; and the third period is either "inefficient combustion" (Line 292) or a combination of the two (Table 3), despite being lower values than either of the other two which are supposedly distinct? If anything, it seems to me that Regime 1 with the one outlier P2 plume should be the mix of burning conditions.

Every plume in Regime 1 had a mean rBC:ΔCO ratio < 0.1. P2 is an outlier; however, it still fits into the classification of inefficient fires. The confusion with Regime 3 was also brought to our attention by the first reviewer, and we have revised the text in Sections 3.1 and 4.1 to explain why the plumes in Regime 3 are likely from a mix of inefficient and efficient despite the overall low rBC:ΔCO ratios.

In Section 3.1, the text now reads, "*The final two plume events, spanning August 24 to September 11, have the lowest overall rBC:ΔCO, with a mean of 0.0071±0.0004. These low rBC:ΔCO values indicate that the fires were inefficient, however, most fires occurred east of 30°E and south of 10°S (Fig. 5a, in northeast Zambia, southwest Tanzania, Mozambique, and Zimbabwe), with surface RH ranging between 30 and 60 % (Fig. 5b), over vegetation types varying from grasslands to woody savannas (Fig. 5c). Also, most of the fires occurred over dry central Africa and many also occurred on the eastern African*

*coast where precipitation was greater in September (Ryoo et al., 2021). The variation in surface RH, and vegetation insinuate that the two plumes in this regime may have originated from fires that are both efficient and inefficient. We further describe these conflicting conditions in Section 4. A notable feature of this regime is that the strong free-tropospheric winds known as the African Easterly Jet-South became active around August 20, at approximately 700 hPa (Ryoo et al., 2022)."*

In Section 4, the text now reads, *"The mean rBC:ΔCO values in Regime 3 were less than 0.01, indicating that the fires across the DRC and Mozambique were inefficient, as concluded above (Figs. 5a, 13c-d). However, the low OA:rBC values, low ΔSO$_4$:rBC values, and high FrBC observed during P9 (late August) suggest that BBA during this plume event originated from efficient fires. A similar discrepancy is seen in P10 (early September), where high OA:rBC, large rBC coating-to-core mass ratios, and high ΔSO$_4$:rBC would suggest that the fires were inefficient, while the high FrBC would indicate that the fires were efficient. These conflicting results imply that despite the overall low rBC:ΔCO values in Regime 3, the BBA from earlier in this regime likely resulted from efficient fires across central African grasslands, whereas the BBA from later in this regime likley resulted from both efficient fires across the grasslands and inefficient fires near the eastern coast (Jiang et al., 2020). These intriguing BBA properties observed in early September are further investigated in Sect. 5.1. Overall, the efficient fires in late August and the combination of efficient and inefficient fires in September are consistent with observations reported by Che et al. (2022a), who concluded that burning conditions in this region become less efficient as cloud cover increases, precipitation increases, surface wind speed decreases, and soil moisture increases from August to October."*

This also gets a bit muddled with the discussion of oxidation and evaporation, transport, and other processes (Section 4, throughout), since it's not clear to start how these plume events are similar to one another. If the transport pathways in Fig 13 vary from t~5 to 9 days in these examples (just from when it exited the continent, it's difficult to follow how this can be used to state that the OA:rBC etc ratios vary at the time of emission, if they're then oxidized at different rates over different times, if I'm following Sec 4. It's not clear how "faster transport pathways (Line ~974+) are definitive in one regime over another. Yes, the reviewer makes a good point here, this is an assumption that we make but did not clearly state. In Regime 1, the OA:rBC ratios are the highest, and transport also took the longest to get to Ascension Island. Whereas in Regime 2, OA:rBC ratios were lower and transport time was the fastest. We assume that OA:rBC values were higher initially due to the inefficient fires and even with 10 days of transport time and that the OA concentrations were still higher when the plume was sampled at Ascension Island. We have added text throughout Section 4.1 to clarify that our interpretation of the OA:rBC ratios measured at Ascension Island are consistent with the burning conditions at the source. We have also revised the following text in the beginning of Section 4.1 to add more transparency to our interpretation.

*"In this section, we discuss how burning conditions and fuel types change across the three temporal regimes, and how these changes may affect BBA properties such as FrBC,*

*rBC coating-to-core mass ratios, and OA:rBC and ΔSO4:rBC mass ratios. These analyses offer further BBA characterization than what has typically been presented for biomass-burning events. However, given the limited data set, we cannot definitively describe conditions over land and instead rely on supporting data from previous studies in this region to interpret the observed patterns."*

Figure 1: caption states "Pink boxes indicate selected plume events; blue boxes indicate selected clean periods" but I'm only seeing grey boxes edged in orange. It might also be nice to label the different plume events on this figure (P1-10).
This has been corrected. We have also labeled the 10 plume events on the figure.

Check that subpanels for Figs 3-5 are labeled properly in the caption and in the text; the caption seems to describe a, c, b rather than a, b, c.
This has been corrected.

A minor note: VIIRS is an instrument on the Suomi NPP satellite (e.g. Fig 3 caption)
This has been corrected. We have also corrected the figure caption to read *"...fires detected by the NASA SUOMI NPP satellite..."* for each of the captions.

Line 51: suggest to report lat/lon in S, W coordinates, rather than negative N,E.
This has been adjusted to S, W coordinates.

Discussion ~Line 47-58: the context might benefit from discussing Eck et al 2013 (doi:10.1002/jgrd.50500), which saw an increase in SSA through the BB season at one site, i.e., changes in optical properties likely from similar geographical regions.
Eck et al., 2013 is a relevant study to this work. We agree with the reviewer that their work should be mentioned in this section. We have edited and added the following text to Section 1, *"The BBA in the FT is highly absorbing of shortwave radiation (Pistone et al., 2019; Denjean et al., 2020a; Taylor et al., 2020; Wu et al., 2020; Dobracki et al., 2023) with single scattering albedo (the ratio of the aerosol scattering coefficient to the sum of the aerosol scattering and absorption coefficients) at wavelength 530 nm ($SSA_{530}$) values increasing from 0.80 to 0.95 between June and October in the continental boundary layer across southern Africa (Eck et al., 2013). Measurements in the FT over the SEA show August-September mean $SSA_{530}$ values near 0.84 (Pistone et al., 2019; Wu et al., 2020, Dobracki et al., 2023) and measurements in the MBL at Ascension Island (7.95 ˚S, 14.36 ˚W), a remote location in the tropical Atlantic, yield an even lower $SSA_{530}$, with June-August monthly-mean values near 0.80 (Zuidema et al., 2018; Che et al., 2022a). The lower $SSA_{530}$ values at Ascension Island in the MBL compared to that in the FT above the island has not been previously explained (Barrett et al., 2022; Sedlacek et al., 2022)."*

Line 181: surely there aren't many fires at 5.7W, 3.2 N? Typo?
These were just the selected box parameters for the NASA FIRMS data selection. We wanted to encompass the entire region. Figures 3c-5c show that there were no fires above 0˚N and of course none over the southeast Atlantic Ocean. We have adjusted the text to

read, *"The locations of the fires between 12.0 ˚W-52.0 ˚E, and 0 ˚N-34.5 ˚S, encompass the sources…"*

Fig 6: I'm curious what the bars are on this plot— is it some standard deviation rather than the percentile distribution over a given event? I ask because in contrast to many of the other figures, these ones seem to be uniform rather than varying from plume to plume, but I would imagine that the range in diameters would vary between plumes as well?
The error bars on the original figure were the standard deviation of the data set. We have updated the figure to include error bars for the standard deviation of each plume.

Sentence starting on Line 205: sentence fragment or missing a verb, I think
We have rewritten the sentence as, *"Oxalate, an organic acid that is a well-known tracer of aqueous-phase OA oxidation contributes to f44 and can indicate that the aerosol has interacted with cloud (Sorooshian et al., 2010; Ervens et al., 2011)."*

Line ~224: rBC units switch back and forth between micrograms and nanograms; I'd pick one. Also it might be good to show the rBC<20ng/m3 threshold in Fig 1, which has 200ng/m3 as the lowest tick mark, if I'm reading it correctly
We have converted the ng m$^{-3}$ to µg m$^{-3}$ in the text and have added blue color bars to Figure 1 to denote clean time periods.

Table 3: I presume these values are means/standard deviations, but it would be good to confirm that in the caption, if you stick with the Regime construction.
We have rewritten the table captions to include "mean ± standard deviation" when relevant.

Table 4: last row might be missing a +/-
This has been corrected.

Fig 8: it might be nice to show the plume events here, as in Fig1, or perhaps as the ratios between parameters as described in the text.
This is a good idea. We have added the colored boxes to the figure to show the plume events. The boxes are also labeled.
Line 479: typo in "mass"
This has been corrected.

Line 584: it's a bit difficult to follow what this sentence is saying.
We have rewritten the sentence as *"Only 4 out of the 12 polluted days in Regime 3 exhibited a bimodal number size distribution (Fig. S4c), which is fewer than in both Regimes 1 and 2. The monomodal number size distributions only contained the accumulation mode, with a number concentrations between 400 and 800 cm$^{-3}$ and a modal diameter near 200 nm, the largest of the three regimes (Fig. 11a)."*

Line 651/Figs 13-14: I'm curious why the 850hPa winds are shown (I think; would be nice to specify in the caption). Especially later in the season, the south African Easterly Jet for continental transport is 700 or 600hPa in Aug or Sep.

The 850 hPa winds are used to demonstrate the boundary layer transport to Ascension Island, as also shown in Haywood et al. (2021), Zhang and Zuidema, (2021) and Che et al. (2022).

Table 5: I'm a bit curious about the ATTO transport pathway that gets African biomass burning smoke to the Amazon sooner than to Ascension in June; is that just based on the season? (I might add that to the header, then). But this is a different value than stated in Fig 16.

The similar transport time of 10 days for the BBA to reach both Ascension Island and Brazil can be attributed to the differences in transport pathways between the MBL and FT. The BBA reaching Ascension Island traveled via slower boundary layer transport, as described in Section 3.3. In contrast, Holanda et al. (2020) reported that BBA sampled at ATTO in Brazil was transported primarily within the FT, facilitated by the African Easterly Jet south, resulting in more rapid transport at higher altitudes. Notably, the lower pollution layer (LPL) in their study corresponds to the lower FT, rather than the MBL. We do agree that this description would be helpful to the reader and provide a clearer explanation as to why the BBA is aged similarly, yet the *f44* values at ATTO were higher than those at Ascension Island. We have added the following text to Section 4.5.

*"The BBA sampled at ATTO was transported primarily within the FT, facilitated by the southern African Easterly Jet, resulting in rapid transport at higher altitudes. As a result, the BBA reached Brazil in a similar time frame to Ascension Island, which likely explains the similarities in f44 and OA:rBC values between the two studies. Notably, the lower pollution layer (LPL) in their study corresponds to the lower FT and not the MBL."*

Line 863: what is the relevance of Arctic observations to the present study? I presume they were also of BB aerosol? But Figure 15 suggests that perhaps the present study and the past Arctic observations were not comparable. I presume the aging/transport time was much longer for Arctic aerosol?

The use of the Arctic data was meant to demonstrate how *f44* and *f60* evolve as fresh BBOA ages. While it is not directly comparable to the African BBOA, it illustrates how different the aged African BBOA is from the fresh Arctic BBOA since there are no studies that report *f44* and *f60* values of fresh African BBOA. We do have language in the text about how the Arctic BBOA is only aged 5 hours and the ORACLES and LASIC BBOA is aged 4-10 days. Figure 15 also highlights that the Arctic BBA is fresh BBA. We have added the following text to Section 4.3, *"Although Arctic BBA likely differs from African BBA in composition, this data is useful to illustrate how the BBA evolves in the initial 0 to 5 hours after emission, for which direct data is not available in our study."*

Line 902 and 946: seem to be conflicted as to whether wet deposition can happen in this region?

The text on line 902 refers specifically to the FT, where precipitation was unlikely to occur. In contrast, the text on line 948 addresses the MBL, where precipitation, though

not a major removal mechanism, is possible due to the presence of marine stratocumulus clouds. Additionally, the text following line 948 highlights that precipitation was not a dominant process in the MBL, as there is no indication of large particles being removed in the aerosol size distributions.

Line 968: missing a word?
We have corrected this sentence. It now reads as, "*These conditions allow more time for interaction with clouds. The highest fraction of activated aerosol particles occurred during this time (Zuidema et al., 2018), consistent with larger accumulation-mode aerosol (Dedrick et al. 2024).*"

Line 1079: wrong verb?
We have corrected this sentence. It now reads as, *"Although these chemical properties are consistent with inefficient combustion, the optical properties are not."*

Fig A1: what averaging is shown in this figure (I don't think this is the 1Hz data)? Also, the caption and the legend seem to be saying two different things re: which line/color is the CAPS?
The data presented in the figure is the 30-minute average of the ARM VAP $SSA_{530}$, the CAPS $SSA_{530}$, and rBC. The caps data were interpolated onto the 30-minute average time scale to match the ARM VAP SSA and rBC values. The ARM VAP $SSA_{530}$ mean for the time period shown is $0.80\pm0.02$ and the CAPS $SSA_{530}$ mean is $0.79\pm0.06$. The mean rBC concentration was $0.55\pm0.2$ to indicate these means were compared during a plume event. The figure caption text has been corrected to agree with the legend on Figure A1.

---

## Author Response (AR2)

We thank the reviewers again for the time and effort that was dedicated to providing these suggestions. The reviewer comments are shown in black, with the author responses shown in blue and any edited manuscript language shown in *italicized blue font.*

Report #2 Response

Page 1065-1070: The SSA530 values in the Ascension Island MBL, between 0.75 and 0.83 (Section 3.2.4), are among the lowest in the world from BBA... In contrast, our results demonstrate that BBA in the MBL is more absorbing than that in the FT.

Remaining question:
The authors conclude that "our results demonstrate that BBA in the MBL is more absorbing than that in the FT".
Authors compared LASIC MBL BBAs (8-10 days?) and ORACLES FT BBAs (4-6 days?) with different transport ages. However, Wu et al. (2020) compared MBL and FT BBAs over Ascension Island with close transport ages. The main distinction here, which leads to the different conclusions is BBAs' transport ages.
I agree that "mixing with clean marine air should increase SSA530 in the MBL", such as in Wu et al. (2020). However, the lower LASIC MBL SSA530 appears to be more associated with longer chemical aging and mass loss than ORACLES FT (such as discussions in Sect. 4.3), which leads to the lower OA/BC rations discussed later in Sect. 4.5.
These may require clearer clarification to avoid potential misleading
Wu et al. (2020) report a mean polluted-BL SSA$_{550}$ of 0.81. Their 'period 1' occurs at the tail end of our P8 and their 'period 3' coincides with our P9. For these 2 plumes, we report a mean SSA530 of ~0.793, with the standard deviation encompassing the mean CLARIFY value. Their Table 1 indicates a higher BC fraction in the BL during their period 3 than period 1, consistent with the slight reduction in SSA between the 2 periods shown in our Fig. 12. Although the SSA variability indicated in our Fig. 12 encompasses the CLARIFY values, the reviewer is correct that the CLARIFY values are likely even higher near the surface, as Wu et al. (2020) document a decrease in SSA with height within the boundary layer. The EXSCALABAR estimates of absorption and extinction are state-of-the-art and will provide a more routinely reliable measurement of SSA than the filter-based measurements applied during the LASIC campaign. The LASIC filter-based absorption measurements compare impressively well to the CAPS-SSA measurements (Fig. A1) while Fig. S6 within Barrett et al. (2020) shows good correspondence between the CAPS absorption and EXSCALABAR PAS absorption, for PM1.0. One explanation for the slightly lower LASIC SSA values than those measured by CLARIFY could be the cut size diameter of the aerosol inlet (Barrett et al. 2020). While the LASIC ARM site used a 1 micron aerodynamic impactor, the FAAM used a 1.3 micron aerodynamic impactor. The particles at the surface that are larger than 1 micron are most likely to be coming from sea spray (see Fig. 11 from the manuscript), which possess an SSA closer to 1.0. This is postulated within Barrett et al. (2022) to explain the differences in the scattering observations between the LASIC ARM site and

the CLARIFY observations, and would be enough to explain the small differences in SSA noted here.

We have added the following text to Section 2.2, lines 159-163 to acknowledge the differences in the impactors used during LASIC and CLARIFY.
*"EXSCALABAR sampled downstream of a 1.3 µm aerodynamic diameter impactor (Taylor et al., 2020), whereas a 1.0 µm aerodynamic impactor was positioned upstream of the instruments deployed in LASIC. Variations in impactor cutoff sizes could introduce biases in the measured aerosol optical properties, particularly the scattering coefficients. Nevertheless, data from both campaigns are utilized in this study to facilitate a comparative analysis."*

We have revised the text to the following, beginning on line 1089, to acknowledge this:

*"This result is surprising, as the mean background $SSA_{530}$ value was 0.98±0.01, and mixing with clean marine air should increase $SSA_{530}$ in the MBL, especially near the surface (Wu et al., 2020). For instance, our mean $SSA_{530}$ value across P8–P9 of 0.79 compares relatively well with the polluted-MBL $SSA_{550}$ value of ~0.81 measured during CLARIFY in the polluted MBL for the same time period. This slight difference can likely be explained by variations in impactor cutoff sizes (Section 2.2; Barrett et al., 2022), where larger aerosol sizes in surface-based measurements contribute to higher SSA values. The slight reduction in SSA from P8 to P9 (Fig. 12) is consistent with the observed increase in rBC fraction from Period 1 to Period 3 during CLARIFY (Wu et al., 2020). Our results further solidify that $SSA_{530}$ is negatively correlated with FrBC (Fig. 17a) and positively correlated with OA:rBC (Fig. 17b) in both the MBL and in the FT between the African plateau and Ascension Island."*

Report #1 Response

There were a couple of additional issues that I noticed which suggest they should ensure the details of the revision are correct:

—The author response regarding comments about the efficient/inefficient classification by both reviewers indicates the authors have moved away from various aspects of this distinction, and I do understand the logic in choosing temporal classifications that are not limited to a specific, somewhat arbitrary calendar month. However, the stated main criterion (cutoff at 0.1 in the rBC:dCO ratio) does not seem to follow that for Regime 2? (Sec 2.2). The third plume has a mean <0.1 (and the range is quite large, besides), so I'm not following how that set can be uniformly considered "efficient," if this is in fact the single criterion being used. They have further clarified that the RH classification should be considered supplementary information only, but that seems a bit inconsistent with the language that says e.g. "RH>50% indicating efficient fires" rather than something less prescriptive like "generally associated with". All together, this distinction seems inconsistent and I think needs to be resolved before publication; the paper may be stronger without trying to fit so strongly these assumptions regarding burn conditions.

We have corrected the text in Section 2.2 to be more prescriptive. The text now reads *"Surface RH fields provided by the NOAA National Center for Environmental Prediction (NCEP) reanalysis are robustly used to assess the burning condition classification. RH values >50% are generally associated with efficient fires and <50% are generally associated with inefficient fires."*

While there is variability in the rBC:ΔCO ratios, the mean values in Regime 2 are overall higher than both Regimes 1 and 3 which suggests that the fires in Regime 2 were more efficient that those in Regimes 1 and 3. Supplementary data sets (i.e., fire location, surface RH, and OA:rBC ratios) also support this classification. We have added the following sentence to Section 2.2 to further emphasize that the rBC:ΔCO is used as a general framework for this paper. *"The rBC:ΔCO classification is used as a general framework in this study, with supplementary context provided by additional factors such as surface RH, land use maps, fire locations, and the OA:rBC and SO4:rBC ratios. This approach is consistent with methodologies commonly used in other studies investigating biomass burning emissions and plume characteristics in this region (Vakkari et al., 2018; Che et al., 2022a)."*

—Figures 3-5 have been revised to include an extra week of observations in order to account for transportation time from fire origin; however, panels b) seem unaltered in terms of color contours and 5a) actually seems to show fewer total fire counts than the initial version, which I don't see how that can be if an extra week of time is added. Regarding the latter, perhaps the authors also meant to change the ending dates for the time period?
The original 5a figure used the J1 VIIRS C1 data, instead of the SUOMI VIIRS C2. I have confirmed that all figures are using SUOMI VIIRS C2 data. All maps are now updated in the manuscript.

[Figure]

—I appreciate the authors clarifying the uncertainty bars on Fig 6, but the new, plume-specific standard deviations seem to be invisible on a few of the plumes; how many measurements went into these statistics and might another metric be more instructive, I wonder?

The uncertainty bars on Fig. 6 represent the standard deviation of the daily mean geometric peak diameters for each plume event. For some plumes, such as Plume 3, the daily mean geometric peak diameter remains consistent across multiple days (e.g., consistently at 125 nm over 7 days), resulting in a standard deviation of zero or near-zero. The time resolution of the rBC data is now included in the caption.

—Typo on Line 934 (Angloa)
This has been corrected.

—I'm still not sure whether Fig 15 (with comparison to data from the Arctic) adds much to the discussion, especially since all the SEA observations will be >>5h old, but I'll leave it to the editor and other reviewer to determine if this is crucial or not.
We have changed Figure 15 to only include ORACLES and LASIC data.

—Line 1322: suggest "for" rather than "from"
We have changed the caption to read as *"SSA$_{530}$ calculated from the PSAP and nephelometer (black)"*.

—I do wish the questions posed at the end of Section 1 had been more explicitly answered/revisited in the conclusion, but I would consider this a more minor/stylistic issue (plus, 5 is a lot for a paper this size).
Given the length of the manuscript, we chose to focus the conclusions on the most useful and interesting takeaways from the study, while addressing each of the posed questions in greater detail within the main body of the manuscript. We believe this approach maintains clarity and ensures that the conclusions highlight the key findings of this work.